



# Estimating 2010–2015 Anthropogenic and Natural Methane Emissions in Canada using ECCC Surface and GOSAT Satellite Observations

Sabour Baray[1], Daniel J. Jacob[2], Joannes D. Maasakkers[3], Jian-Xiong Sheng[4], Melissa P. Sulprizio[2], Dylan B.A. Jones[5], A. Anthony Bloom[6], and Robert McLaren[1]

[1]Centre for Atmospheric Chemistry, York University, Toronto, Canada
[2]Harvard University, Cambridge, MA, USA
[3]SRON Netherlands Institute for Space Research, Utrecht, The Netherlands
[4]Massachusetts Institute of Technology, Cambridge, MA, USA
[5]University of Toronto, Toronto, Canada
[6]Jet Propulsion Laboratory, California Institute of Technology, Pasadena, CA, USA

*Correspondence to*: Sabour Baray (sabour@yorku.ca)

**Abstract.** Methane emissions in Canada have both anthropogenic and natural sources. Anthropogenic emissions are estimated to be 4.1 Tg a$^{-1}$ from 2010–2015 in the Canadian Greenhouse Gas Inventory. Natural emissions, which are mostly due to Boreal wetlands, are the largest methane source in Canada and highly uncertain, on the order of ~20 Tg a$^{-1}$ in biosphere process models. Top-down constraints on Canadian methane emissions using atmospheric observations have been limited by the sparse coverage of both surface and satellite observations. Aircraft studies over the last several years have provided 'snapshot' emissions that have been conflicting with inventory estimates. Here we use surface data from the Environment and Climate Change Canada (ECCC) in situ network and space borne data from the Greenhouse Gases Observing Satellite (GOSAT) to determine 2010–2015 anthropogenic and natural methane emissions in Canada in a Bayesian inverse modelling framework. We use GEOS-Chem to simulate anthropogenic emissions comparable to the Canadian inventory and wetlands emissions using an ensemble of WetCHARTs v1.0 scenarios in addition to other minor natural sources. We conduct a comparative analysis of the monthly natural emissions and yearly anthropogenic emissions optimized by surface and satellite data independently. Mean 2010–2015 posterior emissions using ECCC surface data are 6.0 ± 0.4 Tg a$^{-1}$ for total anthropogenic and 10.5 ± 1.9 Tg a$^{-1}$ for total natural emissions, where the error intervals represent the 1-σ spread in yearly posterior results. These results agree with our posterior using GOSAT data of 6.5 ± 0.7 Tg a$^{-1}$ for total anthropogenic and 11.7 ± 1.2 Tg a$^{-1}$ for total natural emissions. The seasonal pattern of posterior natural emissions using either dataset shows slower to start emissions in the spring and a less intense peak in the summer compared to the mean of WetCHARTs scenarios. We combine ECCC and GOSAT data to evaluate capabilities for sectoral and provincial level inversions and identify limitations. We estimate Energy + Agriculture emissions to be 5.1 ± 1.0 Tg a$^{-1}$ which is 59% higher than the National GHG Inventory. We attribute 39% higher anthropogenic emissions to Western Canada than the prior. Natural emissions are lower across Canada with large downscaling in the Hudson Bay Lowlands. Inversion results are verified against independent aircraft data in Saskatchewan and surface data in Quebec which show better agreement with posterior emissions. This study shows a readjustment of the Canadian methane budget is necessary to better match atmospheric observations with higher anthropogenic emissions partially offset by lower natural emissions.


## 1 Introduction

Methane is a significant greenhouse gas second to carbon dioxide in terms of its direct radiative forcing (Myhre et al., 2013). The mixing ratio of methane has increased from ~720 to ~1800 ppb since pre-industrial times (Hartmann et al., 2013). Present-day global methane emissions are well known to be $550 \pm 60$ Tg a$^{-1}$ (Prather et al., 2012), however recent trends in atmospheric methane since the 1990s are not well understood (Turner et al., 2019). Anthropogenic methane sources include oil and gas activities, livestock, rice cultivation, coal mines, landfills, and wastewater treatment. Natural methane emissions are dominated by wetlands, but also include seeps, termites and biomass burning (Kirschke et al., 2013). The main sink of methane is oxidation by the hydroxyl radical (OH) resulting in a lifetime of $9.1 \pm 0.9$ years (Prather et al., 2012). Improving constraints on national methane emissions is a requirement of mitigation policy (Nisbet et al., 2020). Here we use atmospheric methane observations from the Environment and Climate Change Canada (ECCC) surface network and satellite observations from the Greenhouse Gas Observing Satellite (GOSAT) to estimate Canadian methane emissions and disaggregate anthropogenic and natural sources.

The growth rate of atmospheric methane levelled off from the 1990's to early 2000's. This hiatus continued until 2007 when methane concentrations began a renewed growth continuing to present time (Dlugokencky et al., 2009). Differing hypotheses have attempted to constrain the possible causes of these decadal trends. Associated increases with ethane have attributed recent growth to oil and gas (Hausmann et al., 2016). An increasing trend of isotopically lighter methane has been associated with increasing biogenic emissions from wetlands and agriculture (Nisbet et al., 2016), however decreasing biomass burning emissions may be masking increasing oil and gas emissions in the global isotopic ratios (Worden et al., 2017). Observations of methyl chloroform suggest decreasing OH may have resulted in the renewed growth (Rigby et al., 2017; Turner et al., 2017). Causal attribution of the methane growth rate has continued to be challenging partly because only a 3% source-sink imbalance, or ~20 Tg a$^{-1}$, can result in the observed rate of increase. Hence changes in the relative contributions from anthropogenic and natural sources are key to understanding atmospheric methane.

Atmospheric observations provide constraints on methane emissions. In the Canadian greenhouse gas inventory, anthropogenic emissions are estimated to be 4.1 Tg a$^{-1}$ in 2015 with 68% of emissions originating from the Western Canadian provinces of Alberta (42%), Saskatchewan (17%) and British Columbia (9%). Sectoral contributions over the entire country are from three categories: Energy (49%), Agriculture (29%) and Waste (22%) (Environment and Climate Change Canada, 2017). Natural emissions, which are mostly due to Boreal wetlands, are highly uncertain, on the order of ~10-30 Tg a$^{-1}$ from biosphere process modelling (Miller et al., 2014; Bloom et al., 2017). Studies constraining anthropogenic and/or natural methane emissions within Canada have included the use of surface in situ measurements (Miller et al., 2016; Atherton et al., 2017; Ishiziwa et al., 2019), aircraft campaigns (Johnson et al., 2017; Baray et al., 2018) and satellites (Wecht et al., 2014; Turner et al., 2015; Maasakkers et al., 2020). These observations can determine emissions through mass balance methods or be used in conjunction



with a chemical transport model (CTM). Bayesian inverse modelling constrains prior knowledge of emissions based on the
mismatch between modelled and observed concentrations. This requires reliable mapping of "bottom-up" inventory emissions
for the "top-down" observational constraints to be useful (Jacob et al., 2016). Inverse modelling has been more challenging
for Canada than the United States due to a) the sparsity of surface stations and satellite data (Sheng et al., 2018a), b) a factor
of ~10 lower anthropogenic emissions (Maasakkers et al., 2019), c) large spatially-overlapping emissions from Boreal wetlands
that are highly uncertain (Miller et al., 2014), and d) model biases in the high-latitudes stratosphere (Patra et al., 2011),
compromising interpretation of observed methane columns.

These observing system challenges have made Canadian methane emissions difficult to quantify, however studies have been
showing a consistent story across different scales and measurement platforms. Miller et al. (2014, 2016) determined that the
North American network can successfully constrain Canadian natural emissions and found Boreal wetlands to be lower in
2008 when compared to prior fluxes in the WETCHIMP model. Aircraft campaigns over the Alberta oil and gas sector have
found higher emissions than inventories in the Red Deer and Lloydminster regions (Johnson et al., 2017) and unconventional
oil extraction in the Athabasca Oil Sands region (Baray et al., 2018). Atherton et al. (2017) conducted ground-based mobile
measurements of gas production in British Columbia and determined higher emissions than reported, and Zavala-Araiza et al.
(2018) conducted similar ground-based measurements in Alberta to show a profile of super-emitters dominating the fugitive
methane profile similar to sites in the United States. Ishiziwa et al. (2019) constrained arctic wetlands fluxes to be similar in
magnitude to the mean of the WetCHARTS inventory but with better identified seasonal and interannual variability. Satellite
inversions over North America using the GEOS-Chem CTM and data from SCIAMACHY (Wecht et al., 2014) or GOSAT
(Turner et al., 2015; Maasakkers et al., 2019) consistently require upscaling anthropogenic emissions in Western Canada and
downscaling natural emissions in Boreal Canada to match observations, even with the use of updated Canadian fluxes in
Maasakkers et al. (2019) for anthropogenic (Sheng et al., 2017) and wetlands (Bloom et al., 2017) sources. Inverse modelling
studies that use both in situ and satellite observations are valuable for intercomparison and for identifying the limits of spatial
and temporal discretization that are possible (Lu et al., 2020; Tunnicliffe et al., 2020). The Tropospheric Monitoring Instrument
(TROPOMI) launched in 2017 with a data record beginning in 2018 and is expected to provide significant improvements in
emissions monitoring through denser observational coverage at a similar precision to GOSAT (Hu et al., 2018). It is necessary
to build a reliable historical record of Canadian methane emissions as anthropogenic emissions are sensitive to changes in
policy and economic activity (Rogelj et al., 2018) and natural emissions in Boreal Canada may be sensitive to climate change
(Kirschke et al., 2013).

In this study we use surface observations from the ECCC GHG monitoring network and satellite data from GOSAT to constrain
anthropogenic and natural emissions in Canada. We use the GEOS-Chem CTM to simulate 2010–2015 methane
concentrations. The model setup includes the use of an improved bottom-up inventory for Canadian oil and gas emissions
(Sheng et al., 2017), the WetCHARTS extended ensemble for wetlands emissions (Bloom et al., 2017) and EDGAR v4.3.2 for





other anthropogenic sources. We perform an ensemble forward model analysis which compares six wetlands scenarios to the
ECCC surface observation network to assess the influence of process model configurations on Canadian methane. A series of
Bayesian inverse analyses are performed that use ECCC and GOSAT data independently and in a joint surface-satellite system.
We constrain monthly natural emissions and yearly total anthropogenic emissions from 2010–2015 using ECCC and GOSAT
data independently for intercomparison to produce aggregated-source emissions estimates. We test the limitations of the
ECCC and GOSAT joint observation system towards constraining emissions by inventory sector and according to provincial
boundaries. We demonstrate where the observation system succeeds in providing strong constraints on major emissions sources
and quantify the information content of the system to understand the limitations for resolving all minor Canadian emissions.

## 2 Data and Methods

We use the GEOS-Chem CTM v12-03 (http://acmg.seas.harvard.edu/geos/) to simulate methane fields from 2010–2015 on a
2° x 2.5° global grid and compare to surface observations from the ECCC in situ GHG monitoring network and satellite
observations from GOSAT within the Canadian domain. We test for bias in the global model representation of background
methane using both surface and aircraft in situ data at Canada's most westerly site Estevan Point (ESP) and using global
GOSAT data. The sensitivity of simulated methane in Canada to the use of different wetlands flux parametrization is evaluated
by comparing an ensemble of WetCHARTs v1.0 configurations to ECCC surface observations. The WetCHARTs ensemble
mean along other GEOS-Chem prior emissions are used in the Bayesian inverse analysis which optimizes Canadian sources
using ECCC surface data and GOSAT satellite data independently for comparative analysis. We show the limitations of the
observing system towards subnational level discretization by combining ECCC and GOSAT data in a joint-inversion. Here we
describe the observations, the model, and the inverse analysis in further detail.

### 2.1 Observations

### 2.1.1 In situ Surface Observations

We use continuous measurements from eight sites in the ECCC greenhouse gas monitoring network from 2010–2015. Figure
1 shows a map of the sites and Table 1 provides a descriptive list. The eight sites are Estevan Point, British Columbia (ESP),
Lac La Biche, Alberta (LLB), East Trout Lake, Saskatchewan (ETL), Churchill, Manitoba (CHC), Fraserdale, Ontario (FRA),
Egbert, Ontario (EGB), Chibougamau, Quebec (CHM) and Sable Island, Nova Scotia (SBL). All sites use Picarro cavity ring-
down spectrometers (G1301, G2301 or G2401) measuring dry-air mol fractions of methane with hourly-average precision
better than 1 ppb. For model comparison the measurements are averaged over 4h from 12:00 to 16:00 local time for when the
planetary boundary layer is well-mixed. The instruments are calibrated against World Meteorological Organization (WMO)
certified standard gases. The western most site, ESP, measures methane continuously from a 40 m tower at a lighthouse station
on the west coast of Vancouver Island. ESP is surrounded by forests to the north, east, and south and the Pacific Ocean to the
west. ESP is used to evaluate boundary conditions and model bias in the methane background as it is the least sensitive to





Canadian emissions due to prevailing westerly winds. Sites LLB and ETL are the most sensitive to anthropogenic emissions
in Western Canada. LLB measures continuously from a 50 m tower located in a region of peatlands and forest ~200 km NE
of Edmonton and ~230 km S of Fort McMurray. ETL measures from a height of 105 m located ~150 km north of Prince Albert
surrounded by Boreal forest. The sites in the Hudson Bay Lowlands (HBL) region, CHC and FRA, are the most sensitive to
natural wetlands emissions as this area produces some of the largest methane fluxes in North America. CHC measures
continuously from a 60 m tower in a small port town on the western edge of Hudson Bay surrounded by flat tundra. FRA
measures from a 40 m tower and is located on the southern perimeter of James Bay surrounded by extensive wetlands coverage.
The site CHM in Quebec is also sensitive to natural wetlands emissions and is excluded in the inverse analysis to be used to
verify the posterior results. CHM is substituted by Chapais, Quebec ~50 km away from 2011 onwards. The remaining Central
and Atlantic Canada sites EGB and SBL are sensitive to net outflow from Canadian sources, both natural and urban, and some
emissions from the Eastern United States. EGB is in a small rural village ~80 km north of Toronto and measures from a 25 m
tower.  SBL is on a remote uninhabited island 275 km ESE of Halifax, Nova Scotia and measures from a height of 25 m.

**Table 1:** Descriptive list of ECCC in situ observation sites used in the analysis.

| Site Code | Full Name, Province | Latitude | Longitude | Elevation (asl) / Sampling Height (agl) (m) |
|---|---|---|---|---|
| ESP | Estevan Point, British Columbia | 49.4° N | 126.5° W | 7 / 40 |
| LLB | Lac La Biche, Alberta | 55.0° N | 112.5° W | 548 / 50 |
| ETL | East Trout Lake, Saskatchewan | 54.4° N | 105.0° W | 500 / 105 |
| CHC | Churchill, Manitoba | 58.7° N | 93.8° W | 16 / 60 |
| FRA | Fraserdale, Ontario | 49.8° N | 81.5° W | 210 / 40 |
| EGB | Egbert, Ontario | 44.2° N | 79.8° W | 225 / 25 |
| SBL | Sable Island, Nova Scotia | 43.9° N | 60.0° W | 2 / 25 |
| CHM[*†] | Chibougamau, Quebec | 49.7° N | 74.3° W | 383 / 30 |
| CHA[*†] | Chapais, Quebec | 49.8° N | 75.0° W | 381 / 30 |

*Chibougamau, Quebec is replaced by Chapais, Quebec ~50 km away from 2011 to 2015, overlapping in Fig.1
[†] Site is used to evaluate the posterior inversion results, and is not used in the inversion itself

## 2.1.2 GOSAT Satellite Observations

The Greenhouse Gas Observing Satellite (GOSAT) was launched in January 2009 by the Japan Aerospace Exploration Agency
(JAXA). GOSAT is in a low-Earth polar sun-synchronous orbit with an equator overpass around 13:00 local time. The
TANSO-FTS instrument on-board GOSAT retrieves column-averaged dry air mol fractions of methane using short-wave



infrared (SWIR) solar backscatter in the 1.65 µm absorption band (Butz et al., 2011). Observation pixels in the default mode
are 10 km in diameter separated by 260 km along the orbit track with repeated observations every 3 days. Target mode
observations provide denser spatial coverage over areas of interest. There has been no observed degradation of GOSAT data
quality since the beginning of data collection (Kuze et al., 2016). Here we use version 7 of the University of Leicester proxy
methane retrieval over land from January 2010 to December 2015 (Parker et al., 2011, 2015; ESA CCI GHG project team,
2018). The single-observation precision of GOSAT XCH$_4$ data is 13 ppb, and the relative bias is 2 ppb when validated against
the Total Column Carbon Observing Network (TCCON; Buchwitz et al., 2015). Figure 1 shows the GOSAT observations over
Canada used in our analysis within the domain of 45° N–60° N latitude and 50° W–150° W longitude. The observations used
have passed all quality assurance flags for a total of 45,936 observations from 2010–2015, or approximately ~7600
observations per year. Our analysis excludes glint data over oceans, and cloudy conditions are accounted for by the quality
assurance flags. We avoid using data above 60° N latitude due to higher uncertainty in the satellite retrieval and the model
comparison (Maasakkers et al., 2019; Turner et al., 2015).

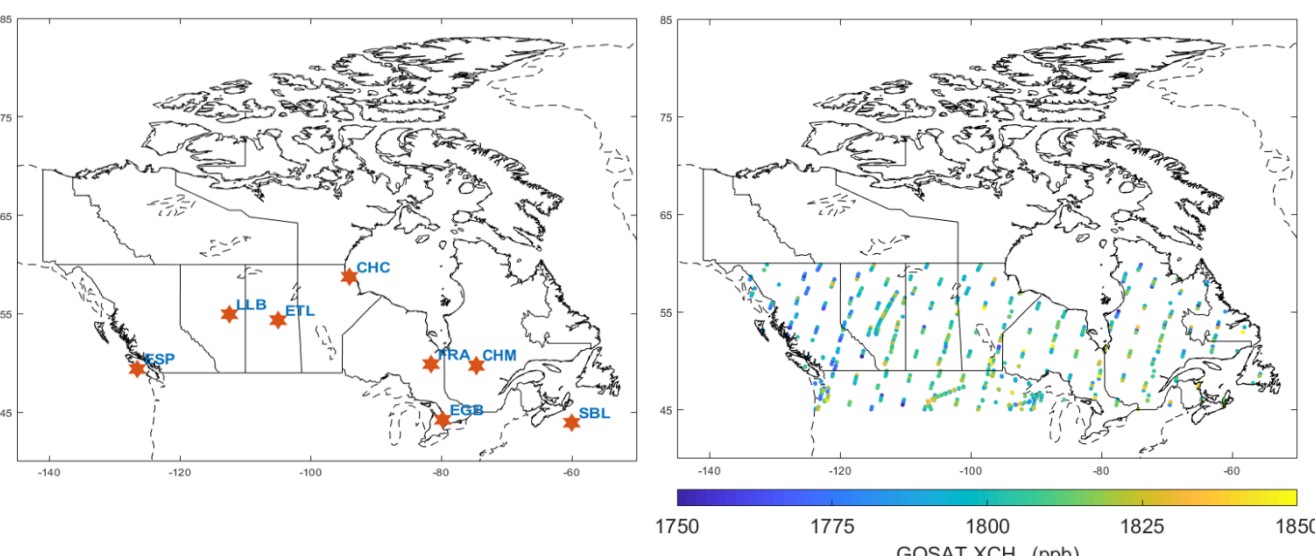


**Figure 1:** ECCC surface (left) and GOSAT satellite (right) observations used in the inverse analysis. A descriptive list of the
ECCC sites is shown in Table 1. GOSAT data shown is from a single year in 2013 and is filtered to the Canadian domain
within 45°N–60°N latitude and 50°W–150°W longitude. There are ~600 GOSAT observations per month in this domain with
a minimum Nov–Jan (112–248) and maximum Jul–Sep (872–1098), individual months are shown in the Supplement (Fig. S1).



## 2.2 Forward Model

We use the GEOS-Chem CTM v12-03 at $2° \times 2.5°$ grid resolution driven by 2009–2015 MERRA-2 meteorological fields from the NASA Global Modeling and Assimilation Office (GMAO). Initial conditions from January 2009 are from a previous GOSAT inversion by Turner et al. (2015) which was shown to be unbiased globally when compared to surface and aircraft data. Bottom-up anthropogenic emissions in GEOS-Chem are from the 2013 ICF Canadian oil and gas inventory (Sheng et al., 2017) and the 2012 EDGAR v4.3.2 global inventory for other Canadian and global sources, and the gridded US 2012 EPA Inventory for the United States (Maasakkers et al., 2016). For wetlands, six configurations from the 2010–2015 extended ensemble of WetCHARTS (Bloom et al., 2017) are used in the ensemble forward model analysis (Section 3.2) and the ensemble mean is used as the prior for the inverse analysis (Sections 3.3–3.4). Figure 2 shows the spatial distribution of the prior methane emissions in Canada from the major anthropogenic and natural sources. The two largest sources are from the ICF oil and gas inventory, (Sheng et al., 2017) and wetlands emissions from the ensemble mean of the WetCHARTS inventory (Bloom et al., 2017), with significant emissions from livestock and waste emissions from EDGAR. Oil and gas are 54% of the anthropogenic total and wetlands are 94% of the natural total. The prior emissions estimates in this simulation are summarized in Table 2, which organizes emissions by Canadian source categories and are compared to sector attribution in the National GHG Inventory (Environment and Climate Change Canada, 2017). Our totals for Energy, Agriculture and Waste are 2.4, 1.0, and 0.9 Tg a$^{-1}$ respectively compared to 2.0, 1.2 and 0.9 Tg a$^{-1}$ in the National Inventory. In the absence of a spatially disaggregated Canadian inventory for methane, we consider these prior estimates reasonably similar for the purpose of comparing our posterior emissions to the National Inventory, however we cannot compare the spatial pattern of emissions which may show less agreement. Emissions from the United States and the rest of the world are included in the model but not optimized in the inversions. Loss of methane from oxidation due to OH is computed using archived 3-D monthly fields of OH from a previous GEOS-Chem full-chemistry simulation (Wecht et al., 2014).





**Table 2:** Mean 2010–2015 prior estimates of Canadian methane emissions used in GEOS-Chem arranged according to categories in the National GHG Emissions Inventory (Environment and Climate Change Canada, 2017).

| Category | | Source Type[a] | Emissions (Tg a⁻¹)[a] | Total (Tg a⁻¹)[a] | Inventory (Tg a⁻¹)[b] |
|---|---|---|---|---|---|
| **Anthropogenic** | **Energy** | Oil | 0.52 | 2.42 | 2.00 |
| | | Gas | 1.81 | | |
| | | Coal | 0.09 | | |
| | **Agriculture** | Livestock | 1.00 | 1.00 | 1.20 |
| | **Waste** | Landfills | 0.66 | 0.94 | 0.92 |
| | | Wastewater | 0.19 | | |
| | | Other Anthropogenic | 0.09 | | |
| **Natural** | **Wetlands** | - | 14.0 | 14.0 | - |
| | **Other Natural** | Biomass Burning | 0.28 | 0.84 | - |
| | | Seeps | 0.28 | | |
| | | Termites | 0.28 | | |

[a]Emissions inputs for GEOS-Chem. These are shown for the individual source types and summed over the categories
Energy, Agriculture and Waste. In Canada, oil and gas are from Sheng et al. (2017), coal, livestock, landfills, wastewater and
other anthropogenic are from EDGAR v4.3.2, wetlands are from Bloom et al. (2017). Biomass burning is from QFED
(Darmenov and da Silva, 2013) and termite emissions are from Fung et al. (1991). Seeps and other global sources are
described in Maasakkers et al. (2019).

[b]Emissions from the National GHG Emissions Inventory (Environment and Climate Change Canada, 2017) that correspond
to the Energy, Agriculture and Waste categories. These are used in the discussion of results but are not included in the
inverse model.



**Figure 2:** Prior estimates of anthropogenic and natural methane emissions. Colour bars are in log scale in units of kg $CH_4$

km$^{-2}$ a$^{-1}$. Most anthropogenic emissions fall under the energy category (A) which are oil and gas in the ICF inventory (Sheng

et al., 2017) plus minor emissions from coal in EDGAR 4.3.2. Livestock (B) and waste (C) are from EDGAR. Natural

emissions are primarily wetlands from the WetCHARTS inventory (D; Bloom et al., 2017).

**2.3 Inverse Model Methodology**

We optimize emissions in the inverse analysis by minimizing the Bayesian cost function $J$ (x) (Rodgers, 2000).

$$J (x) = \tfrac{1}{2} (x - x_a)^T S_a^{-1}(x - x_a) + \tfrac{1}{2} (y - F(x))^T S_o^{-1}(y - F(x)) \qquad (1)$$

Where **x** is the vector of emissions being optimized, **x$_a$** is the vector of prior emissions (Table 2), $F$(x) is the simulation of

methane concentrations corresponding to the observation vector **y** of ECCC surface and/or GOSAT data. **S$_a$** is the prior error





covariance matrix and $\mathbf{S_o}$ is the observational error covariance matrix. The observational error matrix includes both instrument
and model transport error. The GEOS-Chem model relating methane concentrations to emissions $F(x)$ is essentially linear and
can be represented by the Jacobian matrix $\mathbf{K}$ such that $F(x) = \mathbf{K}x + b$, where b is the model background. The background
includes initial conditions from Turner et al. (2015) and methane from global emissions that are held constant in the inversion.
Possible bias in the background is evaluated in detail in Section 3.1 and shown to be minimal. The $\mathbf{K}$ matrix is of $n$ by $m$ size
where $n$ is the number of state vector elements being optimized and $m$ is the number of ECCC surface and/or GOSAT
observations being used. The $\mathbf{K}$ matrix is constructed using the forward mode of GEOS-Chem and the tagged tracer output for
Canadian sources which describes the sensitivity of concentrations to emissions dy/dx in ppb Tg$^{-1}$.

GEOS-Chem continuously simulates global emissions with a global source-sink imbalance of +13 Tg a$^{-1}$ in the budget as
described in Maasakkers et al. (2019). We show in Section 3.1 that this configuration of the model reliably reproduces the
global growth rate in atmospheric methane with adjustments only needed for 2014 and 2015 primarily due to differences in
tropical wetland emissions (Maasakkers et al., 2019). A high resolution inversion over North America over the 2010–2015
time-period using the same prior has shown adjustments to US emissions near the Canadian border are also relatively minimal,
(Maasakkers et al., 2020), so we treat US emissions as constant. This gives a well-represented background for methane which
is checked using global GOSAT data and in situ data at Canadian background site ESP. Hence, we can attribute the model-
observation mismatch $(\mathbf{y} - F(x))$ using observations limited to Canada to Canadian emissions which are optimized in the
inversion. Here we show three inversions with a different number of state vector elements: a) the monthly inversion ($n = 78$)
optimizes monthly natural emissions in Canada and yearly anthropogenic emissions from 2010–2015, b) the sectoral inversion
($n = 5$) optimizes emissions according to the major inventory categories in Table 2 done individually for each year, and c) the
provincial inversion ($n = 16$) optimizes emissions according to subnational boundaries which is also repeated for each year.
The monthly inversion provides high temporal resolution to constrain the seasonality of natural emissions, assuming the spatial
distribution is correct. The sectoral inversion provides direct constraints on inventory categories, and the provincial inversion
provides higher spatial resolution for subnational attribution. Substituting $F(x) = \mathbf{K}x$ in eq. 1 and subtracting the background
b, the analytical solution of the cost function $dJ(x)/dx = 0$ yields the optimal posterior solution $\hat{\mathbf{x}}$ (Rodgers, 2000):

$\hat{x} = x_a + S_a K^T (K S_a K^T + S_o)^{-1} (y - K x_a)$                (2)

The analytical solution provides closed-form error characterization, the posterior error covariance $\hat{\mathbf{S}}$ of the posterior solution
$\hat{\mathbf{x}}$ is given by:

$\hat{S} = (K^T S_o^{-1} K + S_a^{-1})^{-1}$                (3)

The averaging kernel matrix $\mathbf{A}$ is used to evaluate the surface and satellite observing systems and is given by:






$\mathbf{A} = \mathbf{I}_n - \hat{\mathbf{S}}\mathbf{S}_a^{-1}$                                                      (4)

where $\mathbf{I}_n$ is the identity matrix of length $n$ corresponding to the number of state vector elements. The averaging kernel matrix
$\mathbf{A}$ describes the sensitivity of the posterior solution $\hat{\mathbf{x}}$ to the true state x ($\mathbf{A} = d\hat{\mathbf{x}}/d\mathbf{x}$). The trace of $\mathbf{A}$ provides the degrees of
freedom for signal (DOFS), which is the number of pieces of information of the state vector that is gained from the inversion
(DOFS ≤ $n$). The diagonal values of $\mathbf{A}$ provide information on which Canadian state vector elements can be constrained by
ECCC surface and GOSAT satellite observations above the noise, and higher DOFS closer to $n$ correspond to better constrained
sources in total. As a further diagnostic of the inversion we conduct a singular value decomposition of the prewhitened Jacobian
$\check{\mathbf{K}} = \mathbf{S}_o^{-1/2}\mathbf{K}\mathbf{S}_a^{1/2}$ (Rodgers, 2000). The number of singular values greater than one is the effective rank of $\check{\mathbf{K}}$, which shows the
independence of the state vector elements and the number of pieces of information above the noise that are resolved in the
inversion (Heald et al., 2004). The comparison between this eigenanalysis and the DOFS are discussed in the Supplement and
is used to inform the limitations of the observation system.

We construct the prior error covariance matrix $\mathbf{S}_a$ based on aggregated error estimates for source categories and regions. We
use 50% error standard deviation for the aggregated anthropogenic emissions which includes the Sheng et al. (2017) oil and
gas inventory other EDGAR sources, 60% for wetlands emissions from the Bloom et al. (2017) WetCHARTS inventory and
100% for non-wetlands natural sources. We assume no correlation between state vector elements so that $\mathbf{S}_a$ is diagonal.
Anthropogenic emissions have been shown to be spatially uncorrelated (Maasakkers et al., 2016) however wetlands show
spatial correlation (Bloom et al., 2017). Here we optimize broadly aggregated categories, so our method assumes the spatial
pattern of each state vector element is correct, however correlations between state vector elements in the eigenanalysis are
used to assess the limitations of source discretization in the observing systems.

We construct the diagonal observation error matrix $\mathbf{S}_o$ which captures instrument and model error using the relative residual
error method (Heald et al., 2004). In this approach the vector of observed-modelled differences $\Delta = y_{\text{GEOS-Chem}} - y_{\text{observations}}$ is
calculated and the mean observed-modelled difference $\overline{\Delta} = \overline{y_{\text{GEOS-Chem}} - y_{\text{observations}}}$ is attributed to the emissions that will
be optimized. Hence, the standard deviation in the residual error $\Delta' = \Delta - \overline{\Delta}$ represents the observational error and is used as
the diagonal elements of $\mathbf{S}_o$. For our Canadian inversion we find positive model-observation biases in the warmer months
(April to September) and negative biases in the colder months (October to March).We calculate the relative residual error for
growing and non-growing seasons separately, such that $\Delta'$ is partitioned into $\Delta'_g$ (October to March) and $\Delta'_{ng}$ (April to
September) which is then used to calculate the diagonal elements of $\mathbf{S}_o$. For surface observations the mean observational error
is 65 ppb. Since the instrument error is <1 ppb for afternoon mean methane measurements, the observational error is entirely
attributed to transport and representation error of surface methane in the model grid pixels. For satellite observations the mean





observational error is 16 ppb where the instrument error is 11 ppb, showing most of the observational error is from the
instrument rather than the forward model representation of the total column. Column-averaged methane concentrations are
less sensitive to surface emissions resulting in the lower model error (Lu et al., 2020).
**3 Results and Discussion**
**3.1 Evaluation of Bias in the Global Model**
The left panel of Figure 3 shows the comparison of monthly mean GEOS-Chem surface methane concentrations and methane
measured at the ECCC station ESP from 2009 to 2015. ESP is located at the west coast of Vancouver Island (Fig. 1); this site
is used as an evaluation of background methane and tests the bias in the global model as it is the least sensitive to Canadian
emissions due to westerly prevailing winds. The model reliably reproduces surface observations at this station and the growth
rate in background methane due to the source-sink imbalance of +13 Tg a$^{-1}$ in the model global budget (Maasakkers et al.,
2019) with a small mean model-observation bias of 5.3 ppb. The right panel of Figure 3 shows the comparison of modelled
methane to NOAA aircraft profiles at the same site. Aircraft profiles occur approximately once a month continuously over the
study period. The data is not averaged here and is directly compared to GEOS-Chem simulated grid boxes at the pressure level
of the measurement. The reduced mean axis (RMA) regression shows a slope of 0.86 and a coefficient of regression r$^2$ = 0.67
which shows a reasonable model representation of the measurements. These statistics are consistent with previous inversions
using GEOS-Chem that showed relatively unbiased conditions against NOAA surface stations globally (Turner et al., 2015;
Maasakkers et al., 2019). A high resolution inversion over North America over the same 2010–2015 time-period using the
same prior have shown adjustments to US emissions near the Canadian border are relatively minimal (Maasakkers et al., 2020),
so we treat US emissions as constant in the inversion. The acceptable reproducibility of background methane at this site allows
us to attribute much larger differences observed at other sites, up to a maximum of ~1000 ppb in the summer (Figure 6), to
Canadian emissions which are optimized using Canadian observations while holding other global emissions constant.




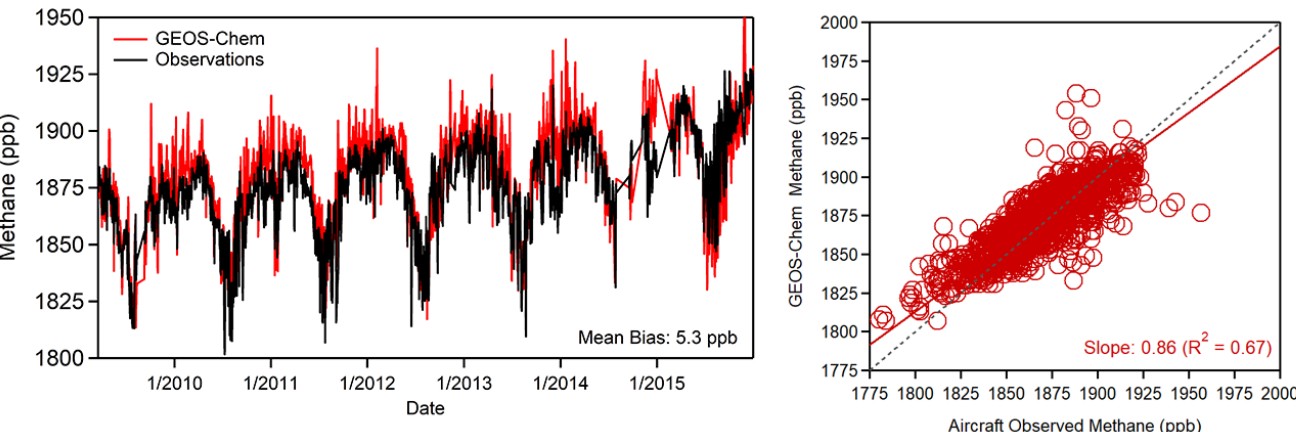



**Figure 3:** Time-series comparison (left) from 2009–2015 of surface GEOS-Chem simulated methane (red) and measured in situ methane (black) at site ESP off the west coast of British Columbia. Comparison to NOAA aircraft profiles (right) from 2009–2015 at the same site using a reduced major axis (RMA) regression along with the 1:1 line (black).


The GEOS-Chem simulation of column averaged methane shows three global biases previously discussed in the literature: (1) a latitude-dependent bias, (2) a seasonal bias and (3) a background change for 2014 and 2015 due to differences in the global source-sink imbalance in these two years (Turner et al., 2015; Saad et al., 2018; Maasakkers et al., 2019; Stanevich et al., 2019). We apply these corrections to the simulated column of methane on a global basis to produce an unbiased background for our target Canadian domain (45° N to 60°N, 50° W to 150° W). The latitude-dependent bias (1) is likely due to excessive polar stratospheric transport (Stanevich et al., 2019). We correct for this bias by fitting the model-GOSAT difference for global $2° \times 2.5°$ grid cells according to a second-order polynomial as shown in Figure 4:

334

$\xi = (2.2\theta^2 - 34\theta) \times 10^{-3} - 2.7$         (5)

336

where $\xi$ is the resulting bias correction in ppb and $\theta$ is latitude in degrees. The correction in this work for the latitude bins of our target domain (45° N to 60° N) is between 0.3 to 2.9 ppb. This correction is lower than what has been shown previously (Turner et al., 2015; Maasakkers et al., 2019) and we attribute this improvement to our use of a 2°x2.5° gridded simulation instead of a 4°x4.5° as recommended by Stanevich et al. (2019) to reduce transport errors. A seasonally oscillating bias (2) remains after this correction. The seasonal bias has an amplitude of ± 4 ppb with repeating maxima in June and minima in December. It is not clear whether this seasonal bias is due to emissions and/or transport errors. In our base case we remove the seasonal bias on a monthly basis following Maasakkers et al. (2019) and show a sensitivity test without the correction for our inversion of monthly natural emissions in Canada (Supplement 1.3). Inversion results using GOSAT data with and without bias corrections in the model simulation of total column methane do not show major differences (Fig. S3). These scenarios all





show agreement with the posterior emissions adjustments determined using ECCC in situ data – which is a useful benchmark
since modelled methane at the surface is not subject to any bias corrections. The background change (3) that appears in the
simulated methane column from 2014 onwards is corrected for in Maasakkers et al. (2019) by optimizing emissions, emissions
trends and trends in OH using a global inversion. In that work correction factors do not appear over Canada and the United
States that would significantly influence the global change in atmospheric methane, and the main adjustment in 2014 and 2015
were to tropical wetlands emissions and OH. Here we treat this as a background change and apply a uniform correction to the
simulated column since emissions outside of Canada and changes in OH are treated as fixed in our Canada-focused inversion.
The background change (3) is 5 ppb in 2014 and 10 ppb in 2015. The right panel of Figure 4 shows the latitude dependent bias
correction and the left panel shows the resulting global time-series of GEOS-Chem total column methane from 2010–2015
after corrections are applied. The global GEOS-Chem – GOSAT differences in the methane column can be limited globally to
within 10 ppb without including the seasonal bias correction, and within 5 ppb with its inclusion. This shows a steady
background in methane for the entire time period from 2010–2015 so global emissions do not affect the optimization of
Canadian emissions. While biases within 10 ppb have been treated as acceptable for methane inversions (Buchwitz et al.,
2015), we evaluate our GOSAT inversion results against inversions with independent ECCC in situ measurements that do not
require any bias corrections in the model (Section 3.3) to produce more robust emissions estimates.

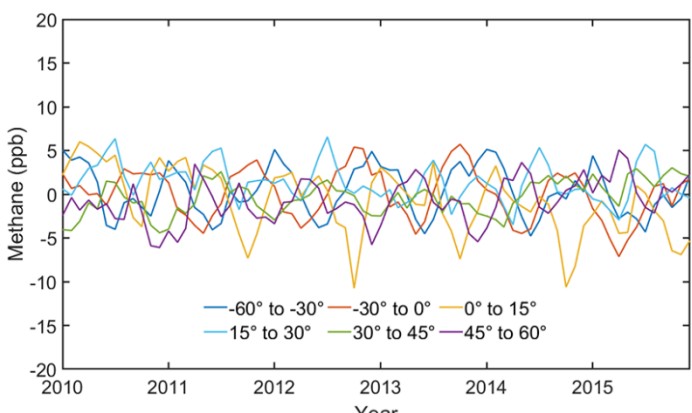
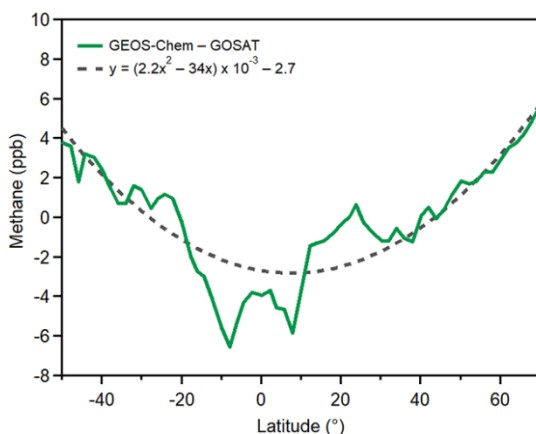


**Figure 4:** Time series (left) from 2010–2015 of the difference between GEOS-Chem simulated total column methane and
GOSAT observations after applying bias corrections, showing a consistent global background for methane. Data used in the
inversion for Canada is from 45° N to 60° N (purple line) and shows acceptable differences within 5 ppb over the entire
global latitude band. To produce the left figure, the latitude-dependent bias (right) is shown with the polynomial correction
that is applied (gray dash) that is within a magnitude of 0.3 to 2.9 ppb for the same latitude.



## 3.2 Evaluation of WetCHARTS Extended Ensemble for Wetlands Emissions in Canada

Wetlands are the largest methane source in Canada with uncertainties in the magnitude, seasonality, and spatial distribution of emissions. Our inverse analysis constrains the magnitude and seasonality of emissions with observations. Ideally, the prior emissions in the model should be the best possible representation of emissions to reduce error in the optimization problem (Jacob et al., 2016). Table 2 shows 2010–2015 mean wetlands emissions in Canada to be 14.0 Tg a$^{-1}$ from the mean of the WetCHARTS v1.0 inventory (Bloom et al., 2017). These emissions are more than three times the total of anthropogenic emissions 4.4 Tg a$^{-1}$. The much larger signal from wetlands emissions poses a difficulty for constraining anthropogenic emissions (Miller et al., 2014). In this section, we evaluate our use of the mean of the WetCHARTS v1.0 extended ensemble by running a series of forward model runs using alternate ensemble members in GEOS-Chem and comparing model output to ECCC in situ observations.

The WetCHARTS extended ensemble for 2010–2015 contains an uncertainty dataset of 18 possible global wetlands configurations as described in Bloom et al. (2017). These depend on three processing parameters which are: three CH$_4$:C temperature-dependent respiration fractions ($q_{10} = 1$, 2, and 3; where 1 is the highest temperature dependency), two inundation extent models (GLWD vs. GLOBCOVER; where GLWD corresponds to higher inundation in Canada) and three global scaling factors for global emissions to amount to 124.5, 166 or 207.5 Tg CH$_4$ yr$^{-1}$ ($3\times2\times3=18$). We find using the scaling factors corresponding to 124.5 and 207.5 Tg CH$_4$ yr$^{-1}$ within GEOS-Chem results in an imbalance in the global budget beyond what is observed in our measurements and degrades the representation of background methane, so we limit the extended ensemble to six members which depend on three temperature parameterizations and two inundation scenarios ($3\times2=6$). Figure 5 shows the magnitude and spatial distribution of wetlands emissions in the six scenarios. The total wetlands emissions within Canada show nearly an order of magnitude difference between ensemble members from 3.9 Tg a$^{-1}$ to 32.4 Tg a$^{-1}$. Compared to the rest of North America, Boreal Canada shows the largest variability between ensemble members, with the Southeast United States as the second most uncertain (Sheng et al., 2018b).

We use ECCC in situ observations to better constrain the range of wetlands methane emissions in the ensemble members. All six configurations are used in GEOS-Chem to produce a series of forward model runs for a subrange of years between 2013–2015. Figure 6 shows GEOS-Chem simulated methane concentrations using the six WetCHARTS configurations and compares to four ECCC in situ measurement sites in Canada (LLB, ETL, FRA, EGB). This subset of available data is representative of sites sensitive to both anthropogenic and natural emissions. Most of Canadian anthropogenic emissions are from Western Canada (Fig. 2), which we use sites LLB and ETL to evaluate (Fig. 1), and a significant amount of Canadian natural emissions are from region surrounding the Hudson's Bay Lowlands, which we use sites FRA and EGB to evaluate. Methane concentrations from GEOS-Chem show large differences when compared to ECCC observations, ranging from +1050 to –150 ppb. The boundary-condition site ESP (Fig. 3) showed a mean bias of 5.3 ppb for all of 2010–2015. Since there is no similar



mismatch in the global representation of methane, these biases up to 1050 ppb can therefore be attributed to misrepresented
local Canadian emissions plus associated transport and representation error. Two types of biases with opposite signs appear
from this comparison. The first type is a positive summertime bias where the modelled methane concentrations significantly
exceed the observations; this bias is more pronounced in sites FRA (Fig. 6-C) and EGB (Fig. 6-D), which are in Ontario and
sensitive to the Hudson Bay Lowlands. The bias is also visible in the western sites LLB (Fig. 6-A) and ETL (Fig. 6-B) to a
lesser extent. As we use a smaller magnitude of wetlands methane emissions corresponding to the ensemble members in Figure
5 (from 32.4 Tg a$^{-1}$ to 3.9 Tg a$^{-1}$), this summertime bias decreases proportionally. Therefore, we can attribute these large
positive summertime biases to growing season wetlands emissions that are overestimated in the process model configurations.
The second type of bias is a year-long negative bias that appears most in site LLB (Fig. 6-A) and is magnified with the use of
lower-magnitude wetlands emissions. This suggests the presence of year-round anthropogenic emissions in Western Canada
that are underestimated in the prior, or that winter-time wetland emissions could also be underestimated in WetCHARTS due
to the lack of explicit soil water and temperature dependencies. The inverse modelling results in the next section attribute this
bias to anthropogenic emissions.

Miller et al. (2016) conducted a study constraining North American Boreal wetlands emissions from the WETCHIMP
inventory modelled in WRF-STILT by comparing to observations in 2008. Their study included the use of three of the ECCC
stations described here (CHM, FRA and ETL). The model comparison to observations in that study showed a similar pattern
of modelled methane exceeding observations in the summer and a low bias at ETL. They suggested wetlands emissions were
overestimated in most model configurations and that the wetlands bias may be masking underestimated anthropogenic
emissions. These conclusions are corroborated by the 2013–2015 comparison shown here, we show high wetlands emissions
configurations in WetCHARTS produce a high bias that exceed measured summertime methane concentrations, and the use
of lower wetlands configurations reveal a year-long low bias apparent in Western Canada. Our results suggest the combined
use of higher inundation extent and lower temperature dependencies (GLWD and $q_{10}$ = 3), or the use of lower inundation
extent and higher temperature dependencies (GLOBCOVER and $q_{10}$ = 1) best reproduce observations near the mean of the
range of emissions, although the ensemble forward model analysis is unable to specify more detailed process model constraints.

The forward model analysis in this section is a direct evaluation of wetlands configurations. This approach allows us *manually*
tune wetlands scenarios and diagnose the sensitivity of the modelled-observed differences to the process modelling parameters.
The inverse analysis shown subsequently is a statistical optimization that applies scaling factors to emissions based on the
same model-observation differences. The inverse analysis can be viewed analogously as an *automatic* approach. These results
show the challenge with optimizing Canadian methane emissions when wetlands emissions are largely uncertain. Our approach
of optimizing anthropogenic and natural emissions simultaneously in an inversion is useful because attempting to constrain
either emissions category, anthropogenic or natural, obfuscates the analysis on the other. We exploit the different pattern of
anthropogenic and natural emissions in time and space (Fig. 6). Natural emissions peak in the summertime and are concentrated





in Boreal Canada, while anthropogenic emissions are persistent year-round and are concentrated in Western Canada (Fig. 2).
Hence when optimizing the model-observation mismatch in a Bayesian inverse framework, some elements of the observation
vector will correspond to high biases from summertime observations in Boreal Canada and some elements will correspond to
low biases in Western Canada. As the choice of prior for the inversion we use the mean of the WetCHARTS configurations
($14.0\ \mathrm{Tg\ a^{-1}}$) which corresponds to the middle of the range shown shaded in red in Figure 6. The 60% range of uncertainty in
the prior error covariance matrix $\mathbf{S}_a$ appropriately excludes the extreme scenarios in Fig. 5 and 6.



**Figure 5:** Ensemble members from the WetCHARTS v1.0 inventory (Bloom et al., 2017) with totals for wetland methane emissions within Canada for each configuration shown in Tg CH$_4$ a$^{-1}$. Ensemble members vary according to the use of three CH$_4$:C q$_{10}$ temperature dependencies and two inundation extent scenarios (GLWD vs. GLOBCOVER) for 3×2=6 scenarios.

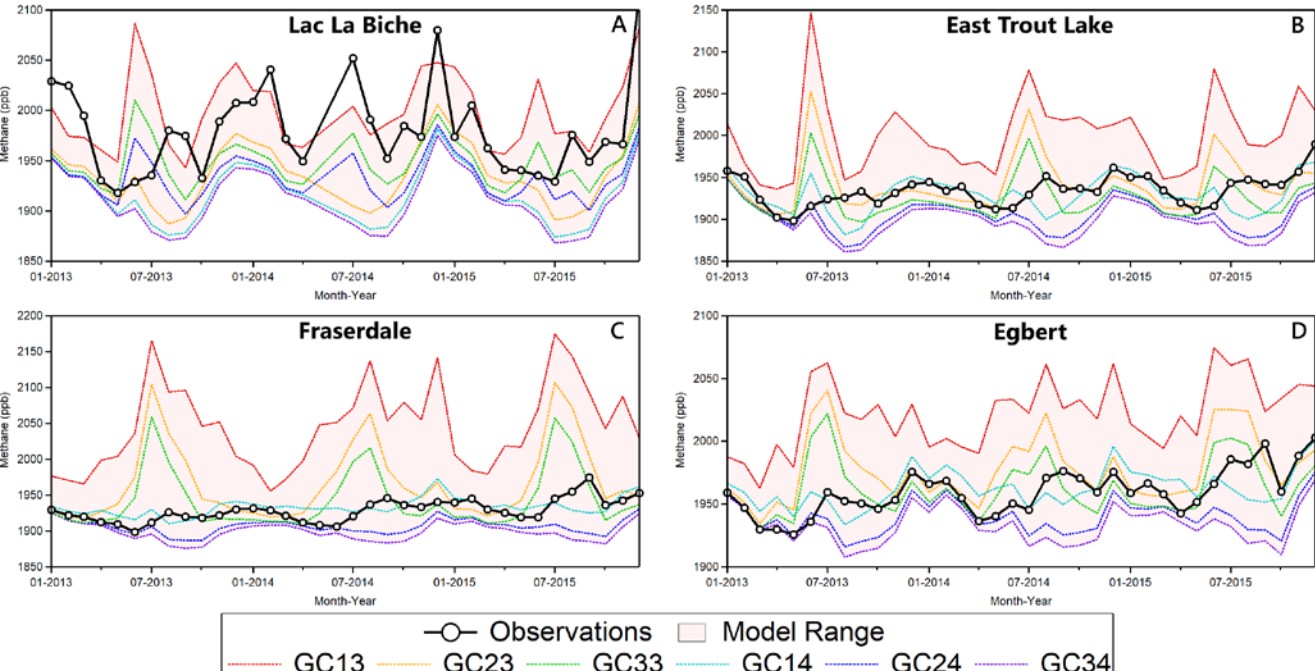

**Figure 6:** Time series of 2013–2015 modelled and observed methane concentrations. Monthly-mean methane from ECCC in situ observations (black) are shown and compared to six GEOS-Chem simulations differing in the use of WetCHARTs ensemble members for wetlands emissions. The six configurations are labelled GCXY where first digit (X=1,2,3) corresponds to the $CH_4$:C $q_{10}$ temperature dependency, which decreases the sensitivity of emissions to temperature with increasing value. The second digit (Y=3,4) corresponds to the model used for inundation extent (3 = GLWD, 4 = GLOBCOVER) where GLOBCOVER produces lower emissions in Canada. Emissions configurations are those shown in Fig. 5 in order of magnitude from red to purple lines, with the shaded red showing the range of concentrations. Sites are LLB, Alberta (A), ETL, Saskatchewan (B), FRA, Northern Ontario (C) and EGB, Southern Ontario (D).



### 3.3 Comparative analysis of inversions using ECCC in situ and GOSAT satellite data

We optimize 2010–2015 emissions in Canada using an $n = 78$ state vector element inversion setup with GOSAT and ECCC
data independently. Elements 1–72 of the inversion are monthly total natural emissions (wetlands + other natural) from 2010–
2015 and elements 73–78 are yearly total anthropogenic emissions (energy + agriculture + waste) for the same years. These
categories correspond to the emissions shown in Table 2. We do not optimize emissions according to clustered grid boxes like
other satellite inversions using GEOS-Chem (Wecht et al., 2014; Turner et al., 2015; Maasakkers et al., 2019) and instead
scale the amplitudes of these two aggregated categories. This approach is a trade-off of time for space, giving up finer spatial
resolution for finer temporal resolution. This is useful for optimizing Canadian methane emissions since a) anthropogenic
emissions are largely concentrated in Western Canada and require less spatial discretization over the entire country and b)
natural emissions are the largest source and have an uncertain seasonality – as shown in the previous section – and require
finer temporal discretization. The limitations of this method are that natural emissions are very unlikely to be spatially
homogenous and vary due to hydrological differences even at the microtopographic level (Bubier et al., 1993). Perfectly
resolving Canadian emissions sources in time and space is challenged by the sparsity and precision of the observing system
and the model representation of the observations. We show the limitations of the combined ECCC and GOSAT observing
system towards resolving subnational emissions in more detail in the subsequent section.

Figure 7 (top) shows 2010-2015 posterior emissions using this 78 state vector approach with ECCC in situ data (blue) and
GOSAT satellite data (green). Error bars are from the diagonal elements of the posterior error covariance matrix $\hat{\mathbf{S}}$. Posterior
anthropogenic emissions averaged over the 6 year period are $6.0 \pm 0.4$ Tg a$^{-1}$ ($1\sigma$ year-to-year variability) using ECCC data
and $6.5 \pm 0.7$ Tg a$^{-1}$ using GOSAT data. Posterior estimates are 36% and 48% higher than the prior of 4.4 Tg a$^{-1}$ for ECCC
and GOSAT results, respectively. There does not appear to be a significant year-to-year trend above the noise for the
anthropogenic emissions optimized by either dataset. The posterior anthropogenic emissions using ECCC and GOSAT data
show agreement with each other in each year but 2011, where the GOSAT derived emissions are statistically higher. The error
from the diagonal of the posterior error covariance matrix $\hat{\mathbf{S}}$ may be overly optimistic, particularly for GOSAT data. This is
due to the observational error covariance matrix $\mathbf{S}_o$ being treated as diagonal when realistically there are correlations between
GOSAT observations that are difficult to quantify (Heald et al., 2004). Our results for anthropogenic emissions show agreement
with top-down aircraft estimates of methane emissions in Alberta that are higher than bottom-up inventories (Johnson et al.,
2017; Baray et al., 2018) and previous satellite inverse-modelling studies over North America that upscale emissions in
Western Canada (Turner et al., 2015; Maasakkers et al., 2019; Maasakkers et al., 2020; Lu et al., 2020). We show source
attribution through a sectoral and subnational scale analysis of anthropogenic emissions in the subsequent section.

Inversion results for monthly natural emissions from 2010–2015 are also shown in Figure 7 (bottom). The total of posterior
natural emissions averaged over the 6 year period is $10.5 \pm 1.9$ Tg a$^{-1}$ using ECCC data and $11.7 \pm 1.2$ Tg a$^{-1}$ using GOSAT





data. The prior for natural emissions is 14.8 Tg a$^{-1}$ from the mean of the WetCHARTS extended ensemble (14.0 Tg a$^{-1}$) plus
other natural (biomass burning + termites + seeps = 0.8 Tg a$^{-1}$). There is some interannual variability in the prior due to higher
emissions in 2010 and 2015. Posterior results averaged over the six years are 29% lower than the prior using ECCC data and
21% lower using GOSAT data, with both posterior results showing agreement with each other. These results are within the
uncertainty range of the WetCHARTS extended ensemble, and we show the magnitude of emissions from the larger uncertainty
dataset (3.9 to 32.4 Tg a$^{-1}$) can be better constrained with both ECCC and GOSAT observations. While our results show lower
natural emissions in all years, a linear fit to the posterior annual emissions using ECCC data shows a trend of increasing natural
emissions at a rate of ~1.0 Tg a$^{-1}$ per year from 2010–2015. The posterior with GOSAT data does not corroborate this result,
the overall emissions trend using GOSAT data is not robust and shows a decreasing trend of ~0.2 Tg a$^{-1}$ per year. The lack of
corroboration of trends between ECCC and GOSAT data may be reflective of the lower overall sensitivity of total column
methane to these surface fluxes (Sheng et al., 2017; Lu et al., 2020) or the inability of this inverse system to constrain trends
sufficiently. Poulter et al. (2017) estimated global wetlands emissions using biogeochemical process models constrained by
inundation and wetlands extend data. They estimated mean annual emissions over all of Boreal North America to be 25.1 ±
11.3 Tg a$^{-1}$ in 2000–2006, 26.1 ± 11.8 Tg a$^{-1}$ in 2007–2012 and 27.1 ± 12.5 Tg a$^{-1}$ which suggests a small increasing trend.
Observational constraints over longer timescales are necessary to investigate the possibility of trends in Canadian natural
methane emissions. Improvements to the observation network and a better understanding of climate sensitivity in
WetCHARTS are necessary to understand how wetlands methane emissions will evolve in future climates.

Figure 8 shows the 2010–2015 average seasonal pattern of natural emissions in the prior and posterior results. The seasonality
of natural methane emissions in the prior shows a sharp peak in July with a narrow methanogenic growing season. The posterior
with ECCC data shows a peak 1-month later in August in most years instead of July, with lower than prior emissions in the
spring months before the peak (March to May) and similar emissions to the prior in the autumn months after the peak
(September to November). Posterior emissions with GOSAT show a peak in July and corroborates the pattern of slower-to-
begin spring emissions and the lower intensity summer peak seen from the ECCC inversion. The posterior results show the
seasonality of emissions is not symmetrical around the temperature peak in July. August emissions are higher than June,
September emissions are higher than May, and October emissions are higher than April. This pattern around July is present in
the prior emissions from WetCHARTS, however the inversion results constrained by ECCC or GOSAT observations intensify
the relative difference between emissions before and after July. Miller et al. (2016) found a similar seasonal pattern of
emissions in the Hudson Bay Lowlands using an inverse model constrained by 2007–2008 in situ data. They found a less
narrow and less intense peak of summertime emissions with higher autumn over spring emissions. Warwick et al. (2016) used
a forward model and isotopic measurements of $\delta^{13}$C-CH$_4$ and $\delta$D-CH$_4$ from 2005–2009 to show northern wetlands emissions
should peak in August-September with a later spring kick-off and later autumn decline. This is further corroborated by Arctic
methane measurements (Thonat et al., 2017) and high latitude eddy covariance measurements (Peltola et al., 2019; Treat et al.,
2018; Zona et al., 2016) that show a larger contribution from the nongrowing season. Our inverse model results using ECCC





and GOSAT data both show agreement with slower to start emissions in the spring and a less intense summertime peak for Canadian wetlands emissions.

Several mechanisms have been proposed to describe a larger relative contribution from cold season methane emissions. Pickett-Heaps et al. (2011) attributed a delayed spring onset in the HBL to the suppression of emissions by snow cover. The temperature dependency in WetCHARTS is based on surface skin temperature (Bloom et al., 2017), however subsurface soil temperatures may continue to sustain methane emissions while the surface is below freezing. When subsurface soil temperatures are near 0°C, this "zero curtain" period can further continue to release methane for an extended period (Zona et al., 2016). Subsurface soils may remain unfrozen at a depth of 40 cm even until December (Miller et al., 2016). Alternatively, field studies in the 1990's suggested the seasonality of emissions may be more influenced by hydrology than temperature, with large differences between peatlands sites (Moore et al., 1994). The WetCHARTS extended ensemble inundation extent variable is constrained seasonally by precipitation. While this does not directly constrain water table depth and wetland extent it provides an aggregate constraint on hydrological variability (Bloom et al., 2017). We show the mean seasonal pattern of both air temperature and precipitation from climatological measurements in subarctic Canada are similarly asymmetrical about the July peak (Fig. S2 in the Supplement). August is warmer and wetter than June, September is warmer and wetter than May, and October is wetter and warmer than April – with wetness being more persistent into the autumn than air temperature. Our inversion results showing a delayed spring start in the seasonal pattern of natural methane emissions in Canada may suggest a lag in the response of methane emissions to temperature and precipitation. This may be due to lingering subsurface soil temperatures and/or more complex parametrization necessary for hydrology.

The overall agreement between ECCC and GOSAT inversions shows robustness in the results. While the same model, prior emissions and inversion procedure are used for assimilating ECCC and GOSAT data, the two datasets are produced with very different measurement methodologies (in situ vs. remote sensing) and sample different parts of the atmosphere (surface concentrations or the total vertical column). The posterior error intervals shown from $\hat{S}$ reflect assumptions about the treatment of observations and may insufficiently account for correlations, however the comparative analysis provides a useful sensitivity test of the posterior emissions since the datasets reflect different treatment of these assumptions.





562

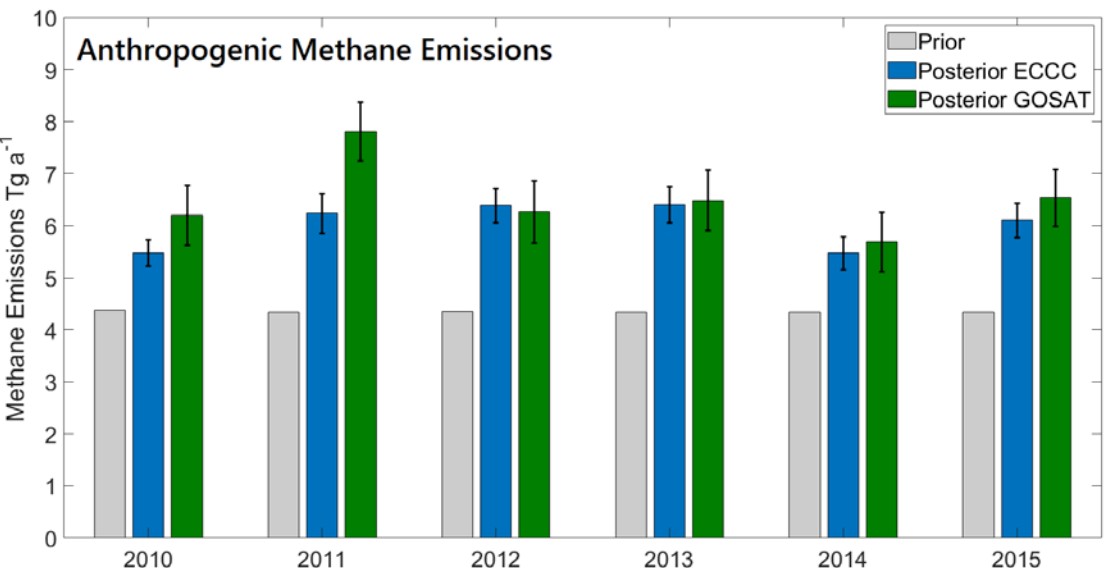

563

**Figure 7:** Comparative analysis of inversion results optimizing annual total Canadian anthropogenic emissions (top) and monthly total natural emissions (bottom) in an $n = 78$ state-vector element setup. The posterior emissions determined using ECCC in situ (blue) and GOSAT satellite (green) data are compared to the prior (gray). Error bars are from the diagonal elements of the posterior error covariance matrix.







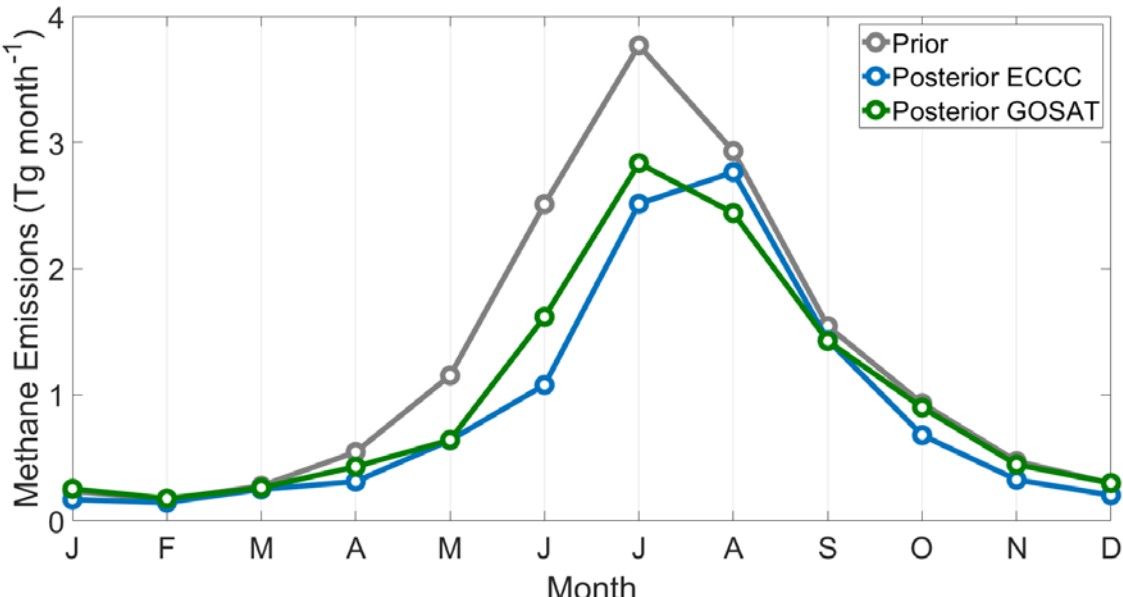


**Figure 8:** Mean 2010–2015 seasonal pattern of natural methane emissions in Tg month$^{-1}$. The annual total emissions are 14.8 Tg a$^{-1}$ (prior, gray), 10.5 ± 1.9 Tg a$^{-1}$ (posterior ECCC, blue) and 11.7 ± 1.2 Tg a$^{-1}$ (posterior GOSAT, green). The posterior results are within the uncertainty range provided by the WetCHARTS extended ensemble (3.9–32.4 Tg a$^{-1}$ for Canada).


### 3.4 Joint-inversions combining ECCC in situ and GOSAT satellite data

We combine the ECCC and GOSAT datasets in two policy-themed inversions: (1) optimizing emissions according to the sectors in the national inventory ($n = 5$ state vector elements; corresponding to the categories in Table 2) and (2) optimizing emissions by provinces split into anthropogenic and natural totals ($n = 16$) and show the results in Figure 9. These inversions are under-determined and show the limitations of the ECCC+GOSAT observing system towards constraining very small magnitude emissions in Canada. We conduct the inversions for each year from 2010–2015 individually and present the average from these six samples. Since these two policy inversions use a low number of state vector elements, they are vulnerable to both aggregation error and overfitting of the well-constrained state vector elements and do not necessarily benefit from using a larger data vector from all six years. We discuss the diagnostics and information content for these inversions in detail in Section 1.4 of the Supplement. The error bars are the 1σ standard deviation of the six yearly results and therefore represent both noise in the inversion procedure and year-to-year differences in the state (emissions and/or transport). Here we do not apply a weighting factor to either dataset, the observations are treated equivalently for the cost function in eq. (1). While there are about 5 times more GOSAT observations than ECCC observations for use in our analysis, the in situ observations have





larger observational error in $S_a$ (due to model error) are much more sensitive to surface fluxes which offset overweighing the
larger amount of GOSAT data. As further diagnostics we show the inversions using GOSAT and ECCC individually (Table
S3 and S4) which show general agreement between the datasets. We also use a singular value decomposition eigenanalysis
(Heald et al., 2004) to evaluate the independence of the state vector elements and to demonstrate which sectoral categories and
provinces can be reliably constrained above the noise in the system  (Fig. S4 and S5 in the Supplement).

Figure 9 (top) shows the sectoral inversion corresponding to categories in the national inventory (Table 2). The prior emissions
with 50% error estimates (60% for wetlands) are 2.4 Tg a$^{-1}$ (Energy), 1.0 Tg a$^{-1}$ (Agriculture), 0.9 Tg a$^{-1}$ (Waste), 14.0 Tg a$^{-1}$
(Wetlands) and 0.8 Tg a$^{-1}$ (Other Natural). The posterior emissions are $3.6 \pm 0.9$ Tg a$^{-1}$ (Energy), $1.5 \pm 0.4$ Tg a$^{-1}$ (Agriculture),
$0.6 \pm 0.3$ Tg a$^{-1}$ (Waste), $9.4 \pm 1.1$ Tg a$^{-1}$ (Wetlands), and $1.7 \pm 0.9$ Tg a$^{-1}$ (Other Natural). The degrees of freedom for signal
and singular value decomposition (Fig. S4) show 3–4 independent pieces of information can be retrieved, which are
differentiated in the figure by solid and hatched bars. The singular value decomposition shows strong source signals
corresponding to wetlands and energy with signal-to-noise ratios of ~37 and ~5, respectively. These are the two largest
emissions sources in Canada and show the inverse system can successfully disentangle the major anthropogenic and natural
contributors. Emissions from waste have a signal-to-noise ratio of ~2 and can be constrained despite the low magnitude of
emissions. This is likely due to waste emissions being more concentrated in Central Canada and away from the influence of
large energy and agriculture emissions in Western Canada. Emissions from other natural sources are at the noise limit and
show a moderate correlation with wetlands, which shows that these two sources are not completely independent. Agriculture
emissions are below the noise in the system and highly correlated with energy emissions. This is likely due to the high spatial
overlap of energy and agriculture emissions in Western Canada. As a result of these limitations, we present the total of energy
and agriculture as $5.1 \pm 1.0$ Tg a$^{-1}$ and the total of wetlands and other natural as $11.1 \pm 1.4$ Tg a$^{-1}$. Our results for total natural
and total anthropogenic emissions are consistent with the results from the previous monthly inversion, with the added benefit
of identifying which sectors are responsible for the higher anthropogenic emissions at the cost of lower temporal resolution.
Waste emissions are 36% lower than the prior and 35% lower than the National GHG Inventory. The total for energy and
agriculture is 49% higher than the prior and 59% higher than the total in the inventory. These results show that energy and/or
agriculture are the sectors that are responsible for the higher anthropogenic emissions.

Figure 9 (bottom) shows the provincial inversion corresponding to the six largest emitting provinces (BC British Columbia,
AB Alberta, SK, Saskatchewan, MB Manitoba, ON Ontario, QC Quebec) and two aggregated regions (ATL Atlantic Canada,
NOR Northern Territories). These regions are further subdivided into total anthropogenic and total natural methane emissions,
with below detection limit anthropogenic emissions from Atlantic Canada and Northern Territories. This inversion especially
challenges the limitations of the ECCC+GOSAT observation system, as only about 8 of 16 independent pieces of information
are retrieved. This means that half of the posterior provincial emissions are below the noise, and we are unable to constrain
province-by-province emissions. The singular value decomposition identifies which regions are well constrained (Fig. S5).





For the anthropogenic emissions AB and ON are strongly constrained. For the natural emissions AB, ON, SK and MB are well
constrained. BC shows correlation between its own anthropogenic and natural emissions and cannot be completely
disaggregated. As a result, we group elements together in Western Canada (BC + AB + SA + MB) and Central Canada (ON +
QC) for interpretation. The total for Western Canada anthropogenic emissions is $4.6 \pm 0.6$ Tg $a^{-1}$ which is 39% higher than the
prior of 3.3 Tg $a^{-1}$. The total for Central Canada is $0.8 \pm 0.2$ Tg $a^{-1}$ which is 11% lower than the prior of 0.9 Tg $a^{-1}$.

Each of our top-down inversion results show higher total anthropogenic emissions than bottom-up estimates. This is consistent
regardless of the observation vector incorporating ECCC data, GOSAT data or ECCC+GOSAT data. The subnational scale
emissions are limited in their ability to provide full characterization of minor emissions across Canada but can successfully
constrain major emissions for source attribution. The sectoral inversion attributes higher anthropogenic emissions to energy
and/or agriculture and applies a small decrease to waste emissions. The provincial inversion attributes higher anthropogenic
emissions to Western Canada and a small decrease to Central Canada. These results suggest that anthropogenic emissions in
Canada are underestimated primarily because of higher emissions from Western Canada energy and/or agriculture. This
interpretation is consistent with previous satellite inverse modelling studies over North America that apply positive scaling
factors to grid box clusters in Western Canada to match observations (Maasakkers et al., 2019; Turner et al., 2015; Wecht et
al., 2014). Aircraft studies in Alberta have also shown higher emissions from oil and gas in Alberta than bottom up estimates
(Baray et al., 2018; Johnson et al., 2017). Atherton et al. (2017) estimated higher emissions from natural gas in north-eastern
British Columbia using mobile surface in situ measurements (Atherton et al., 2017). Zavala-Araiza et al. (2018) showed a
significant amount of methane emissions in Alberta from equipment leaks and venting go unreported due to current reporting
requirements and in some regions a small number of sites may be responsible for most methane emissions. Our inverse
modelling results from 2010–2015 suggest a consistent presence of under-reported or unreported emissions which require a
policy adjustment to reporting practices.












**Figure 9:** Joint-inversions combining 2010–2015 ECCC in situ and GOSAT satellite data showing how the combined observing system remains limited towards resolving all Canadian sources. Inversions are done for each year and we present the six-year average with error bars showing the 1σ standard deviation of the yearly results. Hatched bars indicate sources that are not well-constrained, these are defined as state vector elements with averaging kernel sensitivities less than 0.8 which are





affected by aliasing with other sources (See Supplemental Fig. S4 and S5). The top panel shows the sectoral inversion
according to the categories in the National GHG inventory (Energy, Agriculture, Waste) and two natural categories (Wetlands
and Other Natural). As an example, the diagnostics in Figure S4 shows Agriculture emissions are beneath the noise and cannot
be distinguished from Energy. The bottom panel shows the subnational regional inversion according to provinces (BC British
Columbia, AB Alberta, SK, Saskatchewan, MB Manitoba, ON Ontario, QC Quebec) and aggregated regions (ATL Atlantic
Canada, NOR Northern Territories) further subdivided according to total anthropogenic and total natural emissions. The
diagnostics in Fig. S5 show more than half of the regions are at or below the noise. For anthropogenic emissions, the best
constraints are on provinces AB and ON. For natural emissions, the best constraints are on AB, SK, MB and ON.

### 3.5 Comparison to Independent Aircraft and In situ Data

We test the robustness of the optimized emissions from each of the three inversions shown (monthly natural, sectoral, and
provincial) by comparing to independent measurements not used in the inversions. Prior and posterior simulated methane
concentrations are compared to measurements from NOAA ESRL aircraft profiles at East Trout Lake, Saskatchewan (Mund
et al., 2017) and ECCC surface measurements in sites Chapais and Chibougamau in Quebec, Canada. The surface data was
averaged to daily afternoon means (12:00 to 16:00 local time) in the same manner as the surface measurements used in the
inversion. Aircraft data from the NOAA ESRL profiles coincide spatially with the surface measurements at ETL through a
joint analysis program with Environment and Climate Change Canada and have occurred on a regular basis approximately
once a month from 2005 until present time. Aircraft measurements reach ~7000 m above the surface with samples at multiple
altitudes accomplished using a programmable multi-flask system that is further discussed in Mund et al. (2017), however we
limit the comparison to the lowest 1 km above ground since higher altitude measurements are mostly background. The aircraft
data is not averaged however the flights occur around the same time in the early afternoon.

Figure 10 shows the comparison using reduced-major axis (RMA) regressions with the coefficient of determination ($R^2$), the
slope and the mean-bias shown as metrics to evaluate the agreement. Surface data in CHA, Quebec shows better posterior
agreement with observations according to all metrics for each of the three inversions. The $R^2$ of the prior is 0.36 and improves
to a range of 0.44–0.52 for the posterior results, the slope is 1.17 in the prior and improves to a range of 0.91–1.13 and the
mean bias is –16.4 ppb in the prior and improves to –11.4 and –4.9 ppb. Since this site in Quebec is particularly sensitive to
the Hudson Bay Lowlands, the agreement in all metrics suggests our posterior emissions can better represent wetlands
emissions in this region. This includes the reduced peak seasonality of natural emissions in the monthly inversion, the reduction
of wetlands emissions in the sectoral inversion or the reduction of natural emissions primarily in Central Canada in the
provincial inversion. Aircraft data in Saskatchewan shows improvement in the $R^2$ and mean bias metrics but slightly degrades
the slope in one case. The $R^2$ of the prior is 0.14 and improves to a range of 0.20–0.33, the mean bias of the prior is –6.8ppb
and improves to –0.4 and –1.4 ppb. The slope of the prior is 1.15 which slightly degrades to 0.83 in the monthly inversion and
improves to a range of 0.86–0.91 in the provincial and sectoral inversions. The high resolution aircraft measurements are more



susceptible to representation error at this 2°x2.5° grid resolution. Furthermore, the time-series comparison to surface data at
East Trout Lake (Fig. 6) shows overall lower sensitivity to summertime wetlands emissions than Fraserdale and Egbert, and
lower sensitivity to anthropogenic emissions from Alberta than Lac La Biche. Hence the modelled methane concentrations at
the aircraft measurement points are adjusted less by the change in posterior emissions. However, improvement in the $R^2$ and
mean bias metrics show there is still a better representation of the variance in the data which suggests the posterior emissions
reduce bias due to peak emission episodes.





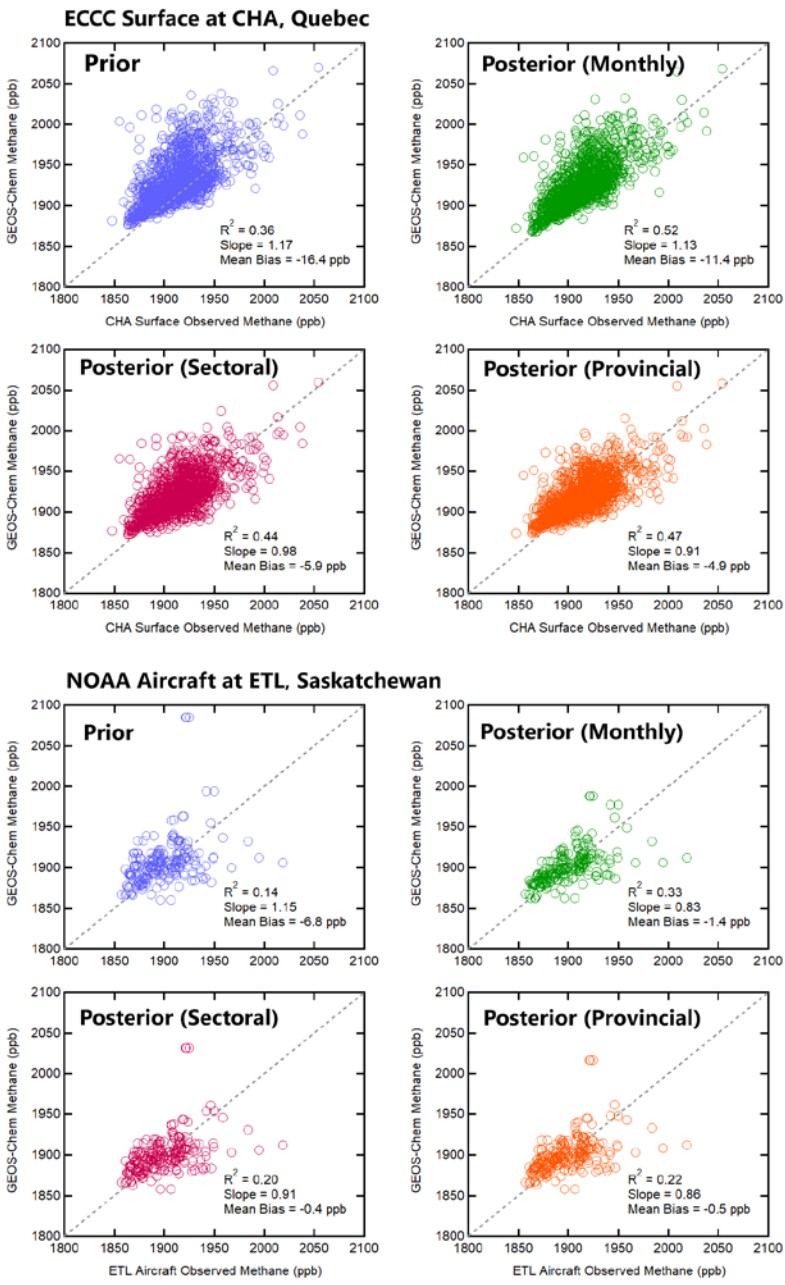

**Figure 10:** Evaluation of inversion results with reduced-major axis (RMA) regressions using independent data. The top four panels show the comparison to ECCC surface observations at Chapais and Chibougamau in Quebec, Canada and the bottom four panels show the comparison to NOAA aircraft profiles at East Trout Lake, Saskatchewan. The agreement of observations with prior simulated methane concentrations (blue) are compared to posterior concentrations using optimized emissions from the monthly inversion (green), the sectoral inversion (magenta), and the provincial inversion (orange). The coefficient of determination ($R^2$), slope and mean bias are shown as metrics of agreement.





## 4 Conclusions


We conduct a Bayesian inverse analysis to optimize anthropogenic and natural methane emissions in Canada using 2010–2015
ECCC in situ and GOSAT satellite observations in GEOS-Chem. Methane concentrations are simulated on a 2°x2.5° grid
using recently updated prior emissions inventories for energy and wetlands emissions in Canada. Modelled background
conditions for the Canadian domain are shown to be unbiased in the comparison to surface in situ data at the western most site
in Canada, Estevan point, with agreement within 6 ppb. A forward model analysis shows much larger biases between –100
ppb and +1050 ppb at surface sites throughout Canada demonstrating the presence of misrepresented local emissions. We
show large positive biases (overestimation of emissions) in the summertime are observed at sites sensitive to wetlands
emissions, these biases are reduced by using lower magnitude wetlands emissions scenarios with lower $CH_4$:C temperature
sensitivities or lower inundation extent. We also show the opposite case of negative biases (underestimation of emissions)
observed year-round at sites in Western Canada. The forward model analysis is consistent with the results of the inverse
analysis that reduce emissions from natural sources and increase emissions from anthropogenic sources to minimize the
mismatch between modelled and observed methane.

We show three approaches for using ECCC and GOSAT data towards inverse modelling of Canadian methane emissions.
These approaches differ according to the temporal and spatial resolution of the solution. We show: (1) a high time-resolution
inversion that solves for natural emissions each month from 2010–2015 and anthropogenic emissions as yearly totals, (2) a
sectoral inversion that solves for emissions according to categories in the national inventory, (3) a provincial inversion that
solves for total anthropogenic and natural emissions at the subnational level. The monthly inversion provides information on
the seasonality of natural emissions (which are ~95% wetlands) but does not provide more depth into anthropogenic emissions
beyond yearly scaling. The sectoral inversion provides more information on the categories of anthropogenic emissions that are
misrepresented in the prior but without spatial detail. The provincial inversion provides the highest level of spatial
discretization but is largely underdetermined due to the limitations of the observing system towards characterizing very low
magnitude emissions from smaller contributing provinces.

Inversion results (1) show mean 2010–2015 posterior emissions for total anthropogenic sources in Canada are $6.0 \pm 0.4$ Tg a$^{-1}$
using ECCC data and $6.5 \pm 0.7$ Tg a$^{-1}$ using GOSAT data. Annual mean natural emissions are $10.5 \pm 1.9$ Tg a$^{-1}$ using ECCC
data and $11.7 \pm 1.2$ Tg a$^{-1}$ using GOSAT data. Both inverse modelling estimates are higher than the prior for anthropogenic
emissions 4.4 Tg a$^{-1}$ and lower than the prior for natural emissions 14.8 Tg a$^{-1}$. Inversion results using both datasets show a
change in the seasonal profile of natural methane emissions where emissions are slower to begin in the spring and show a less
intense peak in the summer. The agreement between two datasets assembled with different measurement methodologies that
sample different parts of the atmosphere is a robust result that lends weight to our conclusions. Our results corroborate recent
studies showing a less-intense and less-narrow summertime peak in North American Boreal wetlands emissions with a higher





relative contribution from the cold season (Miller et al., 2016; Zona et al., 2016; Warwick et al., 2016; Thonat et al., 2017;
Treat et al., 2018; Peltola et al., 2019). These top-down studies using atmospheric observations show biosphere process models
can better account for a more complex response to peak surface soil temperatures.

We also conduct combined ECCC+GOSAT inversions that aim to resolve finer resolution emissions corresponding to (2) the
sectors of the national inventory and corresponding to (3) provincial boundaries. These policy-themed inversions challenge
the capabilities of the ECCC+GOSAT observation system and show the system is not capable of resolving many minor
emissions in Canada. The degrees of freedom for signal for these inversions are 3–4 out of 5 state vector elements for the
sectoral inversion and 8 out of 16 for the provincial inversion. The limitation of this inverse approach towards constraining
sectoral or regional scale emissions in Canada is due to the low magnitude of these emissions, their overlapping nature in
concentrated regions, and the sparsity of data available to distinguish them apart. Grouping correlated sectors together, we
determine $5.1 \pm 1.0$ Tg a$^{-1}$ for energy and agriculture which is 59% higher than the inventory, $0.6 \pm 0.3$ Tg a$^{-1}$ for waste which
is 35% lower than the inventory. For provincial emissions, we show Western Canada is $4.6 \pm 0.6$ Tg a$^{-1}$ which is 39% higher
than the prior and Central Canada is $0.8 \pm 0.2$ which is 11% lower. Both regions show lower natural emissions. These results
show that the higher anthropogenic emissions in the posterior results can be attributed to energy and/or agriculture primarily
in Western Canada where most of Canadian anthropogenic emissions are concentrated. Our results are consistent with other
top-down studies that show higher than reported anthropogenic emissions in Western Canada (Wecht et al., 2014; Turner et
al., 2015; Atherton et al., 2017; Johnson et al., 2017; Baray et al., 2018; Maasakkers et al., 2019). This may be due to oil and
gas emissions that are under-reported or unreported due to current reporting requirements (Zavala-Araiza et al., 2018). These
top-down studies show a need for policy readjustment in reporting practices for Canadian anthropogenic methane emissions.

This study shows the value of using complementary surface and satellite datasets in an inverse analysis. We emphasize the
value of comparative analysis using the datasets independently versus as joint inversions, as minor emissions are too low in
magnitude for the observational precision to distinguish finer scale discretization above the noise. The comparative analysis
has the added benefit of evaluating the datasets against each other and the assumptions that are specific to using either surface
or satellite data. The capabilities for combining and intercomparing datasets is expected to improve, with the launch of
Copernicus Sentinel-5p satellite (TROPOMI) in 2017 and continued expansions on in situ observation networks. The ability
for next generation observations to constrain subnational level emissions in Canada will depend on instrument and model
precision, as well as the emissions magnitudes and spatiotemporal overlap of the targets. These technical capabilities should
be weighed alongside policy needs for improved methane monitoring.





**Competing Interests**

The authors declare that they have no conflict of interest.

**Data Availability**

GEOS-Chem is from http://acmg.seas.harvard.edu/geos/ which includes links to all gridded prior emissions and meteorological fields used in this analysis. GOSAT satellite data is from the University of Leicester v7 proxy retrieval is available through the Copernicus Climate Change Service https://climate.copernicus.eu/. ECCC in situ data is available through the World Data Centre for Greenhouse Gases (WDCGG) https://gaw.kishou.go.jp/. NOAA/ESRL aircraft data is from the Global Monitoring Laboratory https://www.esrl.noaa.gov/gmd/ccgg/aircraft/.

**Author Contributions**

SB, DJJ and RM designed the study. SB conducted the simulations and analysis with contributions from JDM, JXS, MPS, and DBAJ. AAB provided WetCHARTs emissions and supporting data. SB and RM wrote the paper with contributions from all authors. RM was responsible for funding acquisition at York U while DJJ acquired funding at Harvard U.

**Acknowledgements**

Work at Harvard was supported by the NASA Carbon Monitoring System. We thank the Japanese Aerospace Exploration Agency (JAXA) responsible for the GOSAT instrument, and the University of Leicester for the retrieval algorithm used in this analysis. Doug Worthy and the Climate Research Division at Environment and Climate Change Canada are responsible for the in situ surface measurements and the NOAA/ESRL program is responsible for the aircraft methane measurements.





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
