# Peer review of "Estimating 2010–2015 Anthropogenic and Natural Methane"

_Atmospheric Chemistry and Physics, 2020_

## Author Comment (AC1)

**General Comments**

We thank the reviewers for taking the time to provide feedback on our manuscript.

The major concern of Reviewer #1 (Dr. Julia Marshall) was the effect of errors in the global model on the inversion of Canadian emissions. They suggested testing the prior and posterior emissions using upwind and downwind observations outside of the Canadian domain. We find this a reasonable suggestion that would add clarity to the paper. As a result, we have expanded the sensitivity tests and included an evaluation of the posterior Canadian emissions on the global model. These figures have been added to *Section 3.1 Evaluation of Bias in the Global Model* which has been altogether moved to the Supplemental, in line with the suggestion from Reviewer #2 to reduce Section 3 for a more concisely written main manuscript.

Reviewer #2 had some concerns about the design of the inverse model, and suggested future studies to use a finer resolution nested grid simulation that would optimize emissions according to a higher number of spatial and temporal state vector elements. This is also a reasonable suggestion that has promise, however in this study we show that the design of the inverse model is primarily limited by the available observation network. For example, in *Section 3.4* even large spatially aggregated neighboring provinces could not be properly distinguished by the observations. We agree a superior design of the inverse model using GEOS-Chem would be to optimize emissions using a nested grid simulation and to invert emissions at better spatial and temporal resolution. However, we feel a more sophisticated model would better suit a more sophisticated observation system. Improvements to the observation network include an expanded ECCC surface network and satellite observations with either higher density (TROPOMI) or higher precision (GOSAT–2) outside of the years of this analysis.

For this study, the design of the inverse model is to suit the following objectives: (1) to address the larger more apparent biases from Canadian anthropogenic and natural emissions, (2) to show a comparative analysis between inversions using surface and satellite observations and (3) to specifically highlight the limitations of the observing system towards Canadian emissions. We show that higher anthropogenic emissions from Western Canada and lower natural emissions from Boreal Canada better match both ECCC and GOSAT observations and is consistent with a growing body of literature on Canadian methane emissions. We have expanded the sensitivity experiments and posterior analysis to show that the simplifications used in this study design do not alter these conclusions.

Please see the responses to individual comments, the revised manuscript will be submitted with track changes.

**Reviewer #1 (Dr. Julia Marshall)**

**Overall Suggestions:**

*The study presents an inversion analysis with a well-established modelling system using both in situ measurements from the ECCC network and GOSAT satellite measurements. The targeted region for analysis is Canada, and only fluxes in this domain are adjusted and only measurements over this domain are assimilated, which may be problematic (see discussion below). Various state vector setups are presented, where the anthropogenic and biogenic fluxes are scaled separately in time (annually or monthly, respectively) and based on anthropogenic sector or province. Due to the limited number of measurements, these latter approaches are rightly judged to be less robust. The study is interesting and relevant, although I have some concerns about the approach.*

*My major concern with this study has to do with the use of a global simulation where only the Canadian emissions are allowed to be optimized, and where only measurements over Canada are assimilated. It is stated that the "initial conditions from January 2009 are from a previous GOSAT inversion by Turner et al. (2015) which was shown to be unbiased globally when compared to surface and aircraft data". Is this only for the initial 3D fields, or are the optimized fluxes also used for the extra-Canada domain? If not, I am concerned that the flux adjustments seen in Canada might actually be the results of errors elsewhere being adjusted the only way the state vector allows.*

The unbiased initial conditions of the model in 2009 are from Turner et al. (2015). The proceeding global emissions from 2010–2015 are from the prior described in Maasakkers et al. (2019). In that study, the analysis showed the background to be "well simulated in the prior estimate" (our emphasis). The prior reasonably reproduced background measurements when compared to NOAA surface flasks ($R^2 = 0.75$), HIPPO III, IV and V aircraft data ($R^2 = 0.82$) and TCCON data ($R^2 = 0.87$) from 2010–2015 – see Figure 3efg from Maasakkers et al. (2019). Their global inversion optimized the model at 4° x 4.5° grid box resolution which added a 0.84% emissions trend (5 Tg a$^{-1}$), the majority of which were tropical sources (Figure 7c in Maasakkers et al, 2019). This improved the representation of global GOSAT data in 2014 and 2015 (see their Figure 3cd). In our study the underestimation of tropical emissions that affects 2014 and 2015 is addressed with a background correction. Stanevich et al. (2020) presented an issue where polar-stratospheric transport errors were larger in the 4° x 4.5° simulation, which for our study may have increased model errors in the Canada-focused inversion. Since Maasakkers et al. (2019) optimizes emissions on a 4° x 4.5° grid, we chose not to use the posterior and instead used their measurement-tested prior on a 2° x 2.5° grid to reduce the effect of transport errors, with a background correction to address the underestimation of emissions (mostly tropical sources) in later years. This simplification – using a background correction rather than a global inversion – still allows the prior comparison to global GOSAT data in our study (Figure 4 in our manuscript) to match the posterior comparison to global GOSAT data in that study. Furthermore, the net difference in CH$_4$ emissions in our posterior results (higher anthropogenic + lower natural emissions) is approximately –1 to –3 Tg a$^{-1}$ for Canada, which has a relatively minimal impact on the global model. The expected result is a global model which does not degrade or improve,

with an improved characterization of Canadian emissions due to the large local biases present (Figure 6).

However, even with this line of reasoning, we agree with the reviewer it would be better to show more testing of this approach, which will add necessary clarity to the manuscript. We are currently adding sensitivity simulations to better quantify the effect of changes in the model background on the Canadian inversion and to minimize this source of error.

*All validation data are within the Canadian domain, but it is not clear that the fit upstream and downstream of these flux increments is equally consistent. This could be easily tested by comparing a forward run of the prior and posterior fluxes to e.g. soundings outside of the optimized domain. If the adjustments lead to a significantly poorer match to these non-assimilated measurements, there may be a problem with this approach. Methane is long-lived, so not only the measurements immediately downwind should be considered. This is the minimum analysis necessary to test if this approach is reasonable.*

This is a reasonable suggestion; these figures will be added to the supplemental.

*Figure 3, and the discussion of the "acceptable reproducibility of background methane": Again, I am not convinced by this argument that these background fields are so accurate that one can reasonable optimize only Canadian fluxes because everything else agrees so well. Yes, the variability is well matched, but a bias of 5.3 ppb is certainly larger than the measurement uncertainty and, more importantly, looking at the bias alone seems to underestimate the difference. In this plot there seems to be an overestimation of the observations in the earlier part of the record, but by the last year the model seems to be underestimating the measured concentrations. If this boundary condition accepted as is, this difference in the trend will be mapped entirely onto the optimized Canadian fluxes, and not the upstream mismatch where the correction belongs. This trend in the model-observation mismatch appears in Figure 6 as well, even though it is only showing a subset of the simulation period (why?), but at least some of this is already apparent in the un-optimized western boundary. (Notably, the trend of the emissions correction that would be needed to correct this error is positive, just as the trend seen in the ECCC inversion for wetland emissions.)*

While the mean background bias of 5.3 ppb is much larger than the measurement uncertainty for ECCC data, it is much smaller than the observational error used in the inversion of ~65 ppb, which is a combination of instrument error and model error. Hence, the inversion is primarily designed to addresses large biases observed over Canada and not finer scale characterization.

However, the reviewer is correct that the positive–to–negative change in the background is large enough to influence the trend in the ECCC inversion since the surface data was accepted as is. In the original manuscript, the model background for the surface data was left uncorrected (since surface pixels would be less affected by a model underestimation of tropical emissions) and the model background for the GOSAT XCH$_4$ comparison was corrected. The purpose of presenting the results in this manner was to represent an alternate approach to the bias corrections in the ECCC and GOSAT intercomparison. However, this is confusing since it superimposes two

different issues (comparison of ECCC and GOSAT inversion + comparison of using and not using a background correction). This is changed in the revised manuscript and the issue is discussed in better detail.

We address this by expanding the sensitivity tests to include ECCC inversions with an alternate approach to the background bias at the surface. The background bias is corrected in the surface data to match the approach in the GOSAT inversion for an apples-to-apples comparison. In these sensitivity tests, we use the two background sites: Estevan Point (ESP; 49.38°N, 126.54°W) and Alert, Nunavut (ALT; 82.45°N, 62.51°W) for a measurement-based approach to diagnosing errors in the model background. We calculate the mean yearly model–measurement difference at these two sites (Table AR1) and subtract these values as a background correction for all the ECCC data used in the inversion observation vector. This minimizes the influence of the background trend on the ECCC inversion, which is due to under-estimated tropical emissions in 2014 and 2015. For ESP, the mean yearly model–measurement difference is between –7 to +6 ppb, and for ALT the mean difference is between –5 to +12 ppb. We note that in both cases negative bias occurs in the year 2015, consistent with our description of the global model underrepresenting tropical emissions in the later years.

**Table AR1:** Mean yearly model–measurement differences at background sites ESP and ALT.

| Year | Mean Model–Measurement Difference (ESP, ppb)[a] | Mean Model–Measurement Difference (ALT, ppb)[b] |
|------|------|------|
| 2010 | 5.0  | 8.8  |
| 2011 | 5.8  | 8.5  |
| 2012 | 3.6  | 5.9  |
| 2013 | 2.6  | 10.5 |
| 2014 | 2.1  | 11.3 |
| 2015 | -6.9 | -4.7 |

[a]Site ESP is located at 49.38°N, 126.54°W
[b]Site ALT is located at 82.45°N, 62.51°W

Figure AR1 shows the sensitivity inversions comparing the unadjusted ECCC data to the two background-adjusted ECCC inversions using either the mean yearly bias from ESP or ALT. The three inversions are consistent with each other within their error intervals but the adjusted ECCC inversions show improved agreeement with the GOSAT results. For anthropogenic sources, the mean yearly emissions are $6.0 \pm 0.4$ Tg a$^{-1}$ in the unadjusted ECCC inversion, $6.1 \pm 0.4$ Tg a$^{-1}$ with the ESP-adjusted ECCC inversion, and $6.0 \pm 0.4$ Tg a$^{-1}$ with the ALT-adjusted inversion. For natural sources, the mean yearly emissions are $10.5 \pm 1.9$ Tg a$^{-1}$ in the unadjusted ECCC inversion, $12.0 \pm 1.4$ Tg a$^{-1}$ in the ESP-adjusted ECCC inversion, and $11.0 \pm 1.2$ Tg a$^{-1}$ in the ALT-adjusted ECCC inversion. The background-adjusted inversions show higher natural emissions in the years 2010–2014, and lower natural emissions in 2015 due to the negative

background bias that is removed. The background-adjusted inversions show better agreement with the GOSAT mean yearly natural emissions of $11.7 \pm 1.2$ Tg a$^{-1}$. In addition, the trend in natural emissions over this time period is reduced by 40-45% from 1.0 Tg a$^{-1}$ in the unadjusted inversion to 0.55–0.60 Tg a$^{-1}$ in the adjusted inversions. These results show that the background error does not largely affect our central result regarding the the overall increase in anthropogenic emissions and decrease in natural emissions. Correcting for the model background minimizes the projection of under-estimated tropical emissions onto the Canadian fluxes in the later years, which improves the consistency with the GOSAT inversion and significantly reduces the presence of a large trend that was not corroborated by GOSAT.

Considering this background-adjusted analysis reduces the effect of global model errors, it is more suitable to use the mean of the background-adjusted ECCC inversions as the base case in the main text (referred to as "Posterior ECCC" in the revised manuscript), and move the unadjusted ECCC inversion as a senstiivity test in the supplemental. This is reflected in changes to Figure 7, 8, 9 and 10.

[Figure]

**Figure AR1**: Sensitivity analysis of inversion results depending on the use of background correction for ECCC data. Referred to as the monthly inversion, this approach optimizes annual total Canadian anthropogenic emissions (top) and monthly total natural emissions (bottom) in an $n = 78$ state-vector element setup. The prior emissions (gray) are compared to the posterior results using GOSAT (green), and the posterior using ECCC data with an unadjusted background error (blue), ECCC data using a background adjusted according to the yearly difference at ESP (teal) and ALT (purple) from Table AR1.

*Furthermore, the argument that Maasakkers et al. (2020) showed "relatively minimal" adjustments to US emissions near the Canadian border does not mean that US fluxes from further afield do not affect concentrations measured in Canada. Yes, the winds are generally westerly, but air certainly crosses the border in both directions. Not to mention that the stations Egbert and Sable Island have a great deal of US signal when considering only westerly flow, as they are well south of the 49th parallel.*

To address the concern regarding the influence of US emissions near the Canadian border, we introduce a sensitivity test where these two stations most influenced by cross-border transport, Egbert (EGB) and Sable Island (SBL) are removed from the ECCC inversion. Figure AR2 shows a sensitivity test where EGB and SBL (at latitudes of 44.2°N and 43.9°N, respectively) are removed (note in this case, the background is left un-adjusted to avoid overlap in the issues). The mean of anthropogenic emissions in the inversion without these stations is $6.4 \pm 0.6$ Tg a$^{-1}$, and the mean of natural emissions is $10.9 \pm 1.5$ Tg a$^{-1}$. These results are similar to the posterior from the unadjusted ECCC inversion ($6.0 \pm 0.4$ Tg a$^{-1}$ anthropogenic, $10.5 \pm 1.9$ Tg a$^{-1}$ natural) and the GOSAT inversion ($6.5 \pm 0.7$ Tg a$^{-1}$ anthropogenic, $11.7 \pm 1.2$ Tg a$^{-1}$ natural). This sensitivity test shows that the US signal has a minimal influence on the optimization of the large biases due to Canadian emissions. This sensitivity test is added to the Supplemental.

[Figure]

**Figure AR2**: Sensitivity analysis of inversion results depending on the inclusion of sites EGB and SBL which are sensitive to cross-border transport from the United States. Similar to AR1, the prior emissions (gray) are compared to the posterior results using GOSAT (green), and the posterior using ECCC data including all sites (blue) and ECCC data excluding EGB and SBL (yellow).

*The suggested increase in biogenic fluxes from 2010-2015 from the in-situ network is massive – this is on the order of 10% per year! This would be an extraordinary finding, if it can be substantiated. How might this be tested? Did you consider looking at isotope measurements, for example? Why might this not be seen in the GOSAT-only inversion? Why were the GOSAT and ECCC measurements not combined in this "standard" inversion setup as well (as they were in the "policy-themed" inversions presented in Section 3.4). It seems an obvious natural step to do so, to see if this trend is still apparent.*

In the original text it was stated

*L509-L511: The lack of corroboration of trends between ECCC and GOSAT data may be reflective of the lower overall sensitivity of total column methane to these surface fluxes (Sheng et al., 2017; Lu et al., 2020) or the inability of this inverse system to constrain trends sufficiently.*

Given the above results of the sensitivity tests and the reduction in the trend by 40–45% by using a background correction, the latter part of this statement accounts significantly for this result. This entire section has been reworded to better communicate the influence of the background corrections on the magnitude of the trend. In general, the limitations of the method and the study period of 6 years is insufficient for a complete discussion of trends, and it is not a central focus of this study, but the presented trends from ECCC and GOSAT show better agreement when the background bias corrections are similarly matched.

The ECCC and GOSAT data were not combined in this monthly inversion setup to show a comparison of results from the two datasets. A combined ECCC+GOSAT monthly inversion is added and shown in the supplemental, which is within the two results.

*Once these concerns have been addressed the study would be appropriate for publication, but until the robustness of this "regional adjustment only" approach has been tested against independent measurements upwind and downwind of Canada in forward runs of both the prior and posterior fluxes, the scientific conclusions cannot be considered sufficiently robust.*

Thank you for the helpful and clear feedback to improve the quality of the manuscript.

**Minor/typographic comments:**

*L17: have been conflicting -> conflict*
*L26: slower -> a slower*
*L35: specify anthropogenically-influenced GHG: CO2 is less significant than H2O...*
*L54: because only a 3% source-sink imbalance, -> because a source-sink imbalance of only 3%,*

These lines have been changed.

*L58: Please specify that the "Canadian greenhouse gas inventory" is not just an inventory of some prior integrated over Canada, but rather the government report of emissions submitted to the UNFCCC. This is a bit confusing. It's mentioned in the abstract and fully capitalized, as if it were the proper name: "the Canadian Greenhouse Gas Inventory". But then it's also called the "National GHG Inventory" (also capitalized, also in the abstract), and then here just "the Canadian greenhouse gas inventory". None of these match the title of the actual document, which should be explicitly introduced in the introduction.*

The document has now been explicitly introduced in L34 as the National Inventory Report: Greenhouse Gas Sources and Sinks in Canada, a government report of Canada's emissions submitted to the United Nations Framework Convention on Climate Change (UNFCC). It is thereafter referred to as the National Inventory.

*L73: compromising interpretation -> compromising the interpretation*
*L83: wetlands fluxes -> wetland fluxes*
*L86-87: „an increase in" and "a decrease in" would be clearer than "upscaling" and "downscaling" in this context, which could be interpreted as spatial extrapolation/(dis)aggregation.*
*L93: insert comma after first use of "emissions"*
*L113: insert commas before and after "Estevan Point (ESP)"*
*L126: mol -> mole*
*L127: local time for when -> local time, when the*
*L129: western most -> westernmost*
*L136: I guess you mean the largest methane fluxes from wetlands in North America?*

These corrections have been added.

*Section 2.2: I was surprised to see biomass burning not mentioned explicitly in the text, but only listed in the table. It can have quite a bit of interannual and regional variability. I was also surprised to see that the termite emissions were identical to those of biomass burning (in Canada??), and also geologic seeps. Is this just a coincidence, or were these three small sources just distributed evenly over the three (rather different) prior spatial distributions? Please clarify this, also in the text.*

Biomass burning is heavily aliased by wetlands emissions, and the observation network is not capable of resolving the two methane sources. Emissions for biomass burning is from QFED (Darmenov and da Silva, 2013) and termite emissions are from Fung et al. (1991). Seeps and

other global sources are described in Maasakkers et al. (2019). These are different inventories with different spatial patterns, the magnitudes being similar over Canada is by coincidence.

*L191: A couple concerns here, one minor and one major. Here it is optimistically stated that the spatial pattern of emissions "may" show less agreement: this is almost certainly the case, just from a statistical perspective. The major concern has to do with the use of a global simulation where only the Canadian emissions are allowed to be optimized, and where only measurements over Canada are assimilated, but this is discussed elsewhere.*

L191 "may" changed to "most likely". The major concern is addressed in the previous revisions.

*Figure 2: it seems a mistake that the contiguous US/Greenland is not screened out in panel D (but Alaska is), while it is for the other three panels.*

Fixed in Figure 2.

*L250: remove "done"*
*L259: needs a connecter after the comma (e.g. "such that the", "wherein", "and" …)*
*L281: insert "and" before "other"*
*L293: space missing?*

These corrections have been added.

*L294: Did I understand correctly that the in situ data were averaged over the local afternoon each day, essentially giving just one data point per day per station (as described in line 127)? If so, a mean observational error of 65 ppb seems rather massive! Can this be attributed to a poor representation of the spatial distribution of the fluxes, which is not optimized explicitly? The only way the model can adjust the spatial distribution is by changing the weighting of the various categories.*

Yes, the ECCC data amounts to one data point per day per station. The observational error of 65 ppb is much larger than the instrument error due to model representation error using the 2°x2.5° grid. The goal in this study is to address large-signal biases using a relatively sparse observation system, so this broad-category approach is useful. In future studies, model representation error and the characterization of observational error correlations can be improved on to meet a goal of optimizing finer scale emissions using a superior observational network.

*L374, L376 (and elsewhere – find and replace): change "wetlands emissions" to "wetland emissions"*
*L396: and compares -> and compares them*
*L400: from region -> either "from regions" or "from the region"*

These corrections have been added.

*L406: Is Egbert really sensitive to emissions from the Hudson Bay Lowlands? This surprises me. Fraserdale, sure, and maybe Chibougamau, but Egbert? Out of curiosity: for the simulations*

*shown in Figure 6, were the different WetCHARTS scenarios also used for the US fluxes, or were these fixed? Also, please specify that the anthropogenic and "other natural" fluxes from Table 2 were used in the forward simulations shown in Figure 6 (which I assume to be the case).*

We are able to test this using the output from the tagged tracer simulation. We calculate the maximum ΔCH$_4$ to be ~100 ppb each year in the summer specifically from Canadian wetlands emissions when running the simulation with the mean of WetCHARTS scenarios. This matches the mean of the summertime peaks shown in Figure 6; hence the seasonal pattern is primarily accounted for from the ΔCH$_4$ due to Canadian wetlands and not US wetlands. The different WetCHARTS scenarios shown in Figure 6 are not limited to Canada, so the change caused by variable US wetlands emissions is also included in this figure. We show in the previous sensitivity test that excluding EGB and SBL from the ECCC inversion results in similar posterior emissions, so the effect from the US on the results is minimal.

The caption is changed to specify anthropogenic and other natural fluxes are from Table 2.

*L588-590: I don't understand this sentence entirely, there seems to be words missing. Perhaps you mean: While there are about 5 times more GOSAT observations than ECCC observations for use in the analysis and the in-situ observations have larger observational error in Sa (due to model error), the surface measurements are much more sensitive to surface fluxes, which offsets the weight of the larger amount of GOSAT data. Or something like that?*

This line has been changed to the suggested: "While there are about 5 times more GOSAT observations than ECCC observations for use in the analysis and the in-situ observations have larger observational error in Sa (due to model error), the surface measurements are much more sensitive to surface fluxes, which offsets the weight of the larger amount of GOSAT data".

*L688: should "or" be "and"?*

Corrected.

*In Supplement:*
*P7 L167: out -> our*

Corrected.

*Figures S4 and S5: I wonder if these figures might not be easier for the reader to interpret if they were presented as matrices/surface plots? The amplitude of e.g. the singular vector decomposition in the bottom plots could still be indicated somehow, or even kept as line figures, which would help avoid confusion about the interpretation of dashed lines in the middle and bottom panels of Figure S5.*

Added a more reader-friendly figure.

**Reviewer 2**

**Overall suggestions:**

*I felt that the article introduction could flow better with better connection/continuity among topics. Some of the information in the introduction felt out of place (see specific suggestions below). I would think about what argument you want to make in each paragraph of the introduction and use informative topic sentences to guide the reader through each of these arguments.*

The introduction has been condensed, removing L47–56 on the discussion of trends.

*It seems like background estimation was a difficult and challenging process in this study. I would consider adding a second approach to estimating the background -- either by optimizing global fluxes as part of the inverse model or by using a background constructed using atmospheric observations (instead of a model-based background). Section 3.1 of the manuscript includes a lengthy discussion of the merits of the model-based background and whether it is sufficiently accurate for the task at hand. Instead of this lengthy discussion, a second background estimate might be a better way to succinctly quantify the impacts of background uncertainties on the estimated methane fluxes.*

We employed the use of an alternate observation-adjusted background for the ECCC inversion (discussed in Reviewer 1 comments).

*Several sections of the manuscript are relatively long and wordy, especially section 3. In many cases, I think you could cut or condense some of the written material to yield a leaner, punchier, more concise manuscript.*

*Section 3.1* has been expanded to address reviewer comments and moved to the supplemental. This simplifies Section 3 to be more concise.

*I also have some concerns about the inverse modeling setup. I understand that redesigning the inverse modeling framework would require large numbers of new GEOS-Chem runs; hence, I would strongly urge the authors to revise their inverse modeling setup for future studies (even if not the current study). The inverse modeling simulations used in this study either (1) optimize the temporal distribution of fluxes assuming the spatial distribution of the prior is correct, or (2) optimize the spatial distribution of fluxes assuming that the temporal distribution of the prior is correct. In reality, I think both the spatial and temporal distribution of the prior flux estimate could be improved through inverse modeling, and it would be ideal to design an inverse model that does both. Otherwise, I worry that errors in one could interfere your inferences about the other. Also, I think you would see higher model-data correlations in Fig. 10 if your inverse models had more flexibility to adjust both the spatial and temporal distribution of emissions. In future studies, I would also consider using nested North America GEOS-Chem runs instead of using much coarser 2 x 2.5 resolution global simulations.*

This is a reasonable suggestion that has promise, however in this study we show that the design of the inverse model is primarily limited by the observation network. For example, in *Section 3.4* even large spatially aggregated neighboring provinces could not be properly distinguished. We agree a superior design of the inverse model using GEOS-Chem would be to optimize emissions using a nested grid simulation to invert emissions at better spatial and temporal resolution. However, we feel a more sophisticated model would better suit a more sophisticated observation system. Improvements to the observation network include an expanded ECCC surface network and satellite observations with either higher density (TROPOMI) or higher precision (GOSAT–2) outside of the years of this analysis.

**Specific suggestions:**

The abstract is very long at about 400 words. I would consider making the abstract punchier and more concise.

The abstract has been condensed.

Line 37 "however recent trends in": I think this phrase should be a separate sentence from the previous sentence.

Corrected.

Line 47 - 56: This paragraph feels out of place. It doesn't flow with the previous or subsequent paragraphs, and it is not clear how global atmospheric trends are relevant to the current study on Canada. I would consider cutting this paragraph.

This paragraph has been removed.

Line 58: I wasn't clear how the first and second sentences of this paragraph relate to one another. I would try to find a topic sentence for this paragraph that summarizes the overall objective of this paragraph. You might want to have one paragraph about Canada's emissions inventory and another paragraph about existing studies instead of putting both topics in the same paragraph.

Adjusted wording and separated the paragraphs.

Line 75 "however studies have": I think this should be the beginning of a new sentence.

Adjusted wording.

Line 75 "have been showing": replace with "show"

Corrected.

Lines 75 - 95: The information in this paragraph overlaps with the information in the previous paragraph. I would either come up with a unique topic sentence for this paragraph to differentiate this paragraph from the previous one, or I would combine the discussion of top-down studies in this paragraph with the discussion of top-down studies in the previous paragraph.

Adjusted wording.

Line 105 "intercomparison": Why not use "comparison" instead?

Adjusted wording.

Line 116 "mean along other GEOS-Chem prior emissions": It feels like there is a word missing here.

Adjusted wording.

Line 147: Was the Chibougamau site decommissioned in 2011, or did it come back online into operation after 2015? This distinction isn't clear in the wording of line 147.

Adjusted wording. The Chibougamau site was moved to an alternate location with a new name.

Figure 1: I believe that ECCC has several observation sites in Northwest Territories and Nunavut. Why not include those sites in the inverse model? See the list of ECCC sites shown in Fig. 2.6: https://www.nrcan.gc.ca/sites/www.nrcan.gc.ca/files/energy/Climate-change/pdf/CCCR_FULLREPORT-EN-FINAL.pdf.

The sites included in this study were those that were made available on the public domain for the broader scientific community on the World Data Centre for Greenhouse Gases (WDCGG; https://gaw.kishou.go.jp/). Measurements that were not yet available publicly could not be used.

Equations 1, 2, and 3: I think that vectors should be displayed in bold-italic font and matrices in bold font.

Corrected.

Line 235: Should the dimensions of K be m by n, given the definitions for m and n in the article?

Corrected.

Line 248: What are you optimizing for in the monthly inversion? Are you estimating methane fluxes from each individual model grid box in each month? If that were the case, I think the value of n here would be larger. Or are you optimizing something else?

The inversion is not by grid box due to limitations in the observation network. We are optimizing all aggregated Canadian anthropogenic emissions according to a yearly scaling factor, and all aggregated Canadian natural emissions according to monthly scaling factors. These limitations in

the state vector *n* are ultimately due to the limitations of the surface and satellite observation network, which we explore the limitations of in the combined inversions.

Line 252: I wouldn't refer to a monthly inversion as "high temporal resolution". I have seen existing studies estimate daily methane fluxes in an inverse model, and numerous inverse modeling studies of CO2 estimate 3-hourly fluxes.

Corrected. We rephrase this to the 'higher' temporal resolution and specify it is relative to the other approaches in this study.

Line 474 - 475: I disagree that there's a tradeoff between spatial resolution and temporal resolution in the inverse model. Alternative approaches would be to (1) use the GEOS-Chem adjoint in the inverse model, or (2) use a Lagrangian model like Flexpart or STILT in the inverse model. Those approaches would not necessitate a trade-off between the spatial and temporal resolution of the inverse model.

Adjusted wording. The model itself has the technical capabilities to resolve emissions both spatially and temporally. For the Canadian domain, the limitations are not due to the model but due to the observation network, and the design of the coarse inverse model used in this study is chosen to suit this problem.

Line 582 "magnitude emissions in Canada": Is there a word missing here?

Corrected.

We thank Reviewer 2 for their time. The suggestions have improved the quality of the manuscript.

**References**

Darmenov, A. and da Silva, A.: The quick fire emissions dataset (QFED)–documentation of versions 2.1, 2.2 and 2.4, NASA Technical Report Series on Global Modeling and Data Assimilation, NASA TM-2013-104606, 32, 183 pp., 2013.

Fung, I., John, J., Lerner, J., Matthews, E., Prather, M., Steele, L., and Fraser, P.: Three-dimensional model synthesis of the global methane cycle, J. Geophys. Res.-Atmos., 96, 13033–13065, 1991.

Maasakkers, J. D., Jacob, D. J., Sulprizio, M. P., Scarpelli, T. R., Nesser, H., Sheng, J.-X., Zhang, Y., Hersher, M., Bloom, A. A., Bowman, K. W., Worden, J. R., Janssens-Maenhout, G. and Parker, R. J.: Global distribution of methane emissions, emission trends, and OH concentrations and trends inferred from an inversion of GOSAT satellite data for 2010–2015, Atmos. Chem. Phys., 19(11), 7859–7881, doi:10.5194/acp-19-7859-2019, 2019.

Maasakkers, J. D., Jacob, D. J., Sulprizio, M. P., Scarpelli, T. R., Nesser, H., Sheng, J., Zhang, Y., Lu, X., Bloom, A. A., Bowman, K. W., Worden, J. R., and Parker, R. J.: 2010–2015 North American methane emissions, sectoral contributions, and trends: a high-resolution inversion of GOSAT observations of atmospheric methane, Atmos. Chem. Phys., 21, 4339–4356, https://doi.org/10.5194/acp-21-4339-2021, 2021.

Stanevich, I., Jones, D. B. A., Strong, K., Parker, R. J., Boesch, H., Wunch, D., Notholt, J., Petri, C., Warneke, T., Sussmann, R., Schneider, M., Hase, F., Kivi, R., Deutscher, N. M., Velazco, V. A., Walker, K. A., and Deng, F.: Characterizing model errors in chemical transport modeling of methane: impact of model resolution in versions v9-02 of GEOS-Chem and v35j of its adjoint model, Geosci. Model Dev., 13, 3839–3862, https://doi.org/10.5194/gmd-13-3839-2020, 2020.

Turner, A. J., Jacob, D. J., Wecht, K. J., Maasakkers, J. D., Lundgren, E., Andrews, A. E., Biraud, S. C., Boesch, H., Bowman, K. W., Deutscher, N. M., Dubey, M. K., Griffith, D. W. T., Hase, F., Kuze, A., Notholt, J., Ohyama, H., Parker, R., Payne, V. H., Sussmann, R., Sweeney, C., Velazco, V. A., Warneke, T., Wennberg, P. O. and Wunch, D.: Estimating global and North American methane emissions with high spatial resolution using GOSAT satellite data, Atmos. Chem. Phys., 15(12), 7049–7069, doi:10.5194/acp-15-7049-2015, 2015.

---

## Author Response (AR1)

**General Comments**

We thank the reviewers for taking the time to provide feedback on our manuscript.

The major concern of Reviewer #1 (Dr. Julia Marshall) was the effect of errors in the global model on the inversion of Canadian emissions. They suggested testing the prior and posterior emissions using upwind and downwind observations outside of the Canadian domain. We find this a reasonable suggestion that would add clarity to the paper. As a result, we have expanded the sensitivity tests and included an evaluation of the posterior Canadian emissions on the global model. These figures have been added to *Section 3.1 Evaluation of Bias in the Global Model* which has been altogether moved to the Supplement, in line with the suggestion from Reviewer #2 to reduce Section 3 for a more concisely written main manuscript.

Reviewer #2 had some concerns about the design of the inverse model, and suggested future studies to use a finer resolution nested grid simulation that would optimize emissions according to a higher number of spatial and temporal state vector elements. This is also a reasonable suggestion that has promise, however in this study we show that the design of the inverse model is primarily limited by the available observation network. For example, in *Section 3.4* even large spatially aggregated neighboring provinces could not be properly distinguished by the observations. We agree a superior design of the inverse model using GEOS-Chem would be to optimize emissions using a nested grid simulation and to invert emissions at better spatial and temporal resolution. However, we feel a more sophisticated model would better suit a more sophisticated observation system. Improvements to the observation network include an expanded ECCC surface network and satellite observations with either higher density (TROPOMI) or higher precision (GOSAT–2) outside of the years of this analysis.

For this study, the design of the inverse model is to suit the following objectives: (1) to address the larger more apparent biases from Canadian anthropogenic and natural emissions, (2) to show a comparative analysis between inversions using surface and satellite observations and (3) to specifically highlight the limitations of the observing system towards resolving subnational Canadian emissions. We show that higher anthropogenic emissions from Western Canada and lower natural emissions from Boreal Canada better match both ECCC and GOSAT observations and are consistent with a growing body of literature on Canadian methane emissions. We have expanded the sensitivity experiments and posterior analysis to show that the simplifications used in this study design do not alter these conclusions.

Please see the responses to individual comments, the revised manuscript is attached with track changes.

**Reviewer #1 (Dr. Julia Marshall)**

**Overall Suggestions:**

*The study presents an inversion analysis with a well-established modelling system using both in situ measurements from the ECCC network and GOSAT satellite measurements. The targeted region for analysis is Canada, and only fluxes in this domain are adjusted and only measurements over this domain are assimilated, which may be problematic (see discussion below). Various state vector setups are presented, where the anthropogenic and biogenic fluxes are scaled separately in time (annually or monthly, respectively) and based on anthropogenic sector or province. Due to the limited number of measurements, these latter approaches are rightly judged to be less robust. The study is interesting and relevant, although I have some concerns about the approach.*

*My major concern with this study has to do with the use of a global simulation where only the Canadian emissions are allowed to be optimized, and where only measurements over Canada are assimilated. It is stated that the "initial conditions from January 2009 are from a previous GOSAT inversion by Turner et al. (2015) which was shown to be unbiased globally when compared to surface and aircraft data". Is this only for the initial 3D fields, or are the optimized fluxes also used for the extra-Canada domain? If not, I am concerned that the flux adjustments seen in Canada might actually be the results of errors elsewhere being adjusted the only way the state vector allows.*

We agree with the reviewer it would be better to show more testing of this approach, which will add necessary clarity to the manuscript. We have added several sensitivity tests to better quantify the effect of changes in the model background on the Canadian inversion and to minimize this source of error. These have been added to *Section 1.3 Evaluation of Bias in the Global Model* in the Supplement. These tests are discussed in more detail in the next points. Below, we clarify the line of reasoning behind the regional-only simplification used in this study.

The unbiased initial conditions of the model in 2009 are from Turner et al. (2015). The proceeding global emissions from 2010–2015 are from the prior described in Maasakkers et al. (2019). In that study, the analysis showed the background to be "well simulated in the prior estimate" (our emphasis). The prior reasonably reproduced background measurements when compared to NOAA surface flasks ($R^2 = 0.75$), HIPPO III, IV and V aircraft data ($R^2 = 0.82$) and TCCON data ($R^2 = 0.87$) from 2010–2015 – see Figure 3efg from Maasakkers et al. (2019). Their global inversion optimized the model at 4° x 4.5° grid box resolution which added a 0.84% $a^{-1}$ emissions trend (5 Tg $a^{-1}$), the majority of which were tropical sources (Figure 7c in Maasakkers et al, 2019). This improved the representation of global GOSAT data in 2014 and 2015 (see their Figure 3cd). In our study the underestimation of tropical emissions that affects 2014 and 2015 is addressed with a background correction. Stanevich et al. (2020) presented an issue where polar-stratospheric transport errors were larger in the 4° x 4.5° simulation, which for our study may have increased model errors in the Canada-focused inversion. Since Maasakkers et al. (2019) optimizes emissions on a 4° x 4.5° grid, we chose not to use the posterior and instead used their measurement-tested prior on a 2° x 2.5° grid to reduce the effect of transport errors, with a background correction to address the underestimation of emissions (mostly tropical

sources) in later years. This simplification – using a background correction rather than a global inversion – still allows the prior comparison to global GOSAT data in our study (Figure S6 in the revised manuscript) to match the posterior comparison to global GOSAT data in that study. Furthermore, the net difference in $CH_4$ emissions in our posterior results (higher anthropogenic + lower natural emissions) is approximately –1 to –3 Tg a$^{-1}$ for Canada, which has a relatively minimal impact on the global model. The expected result is a global model which does not degrade or improve (new Figure S8), with an improved characterization of Canadian emissions (Figure 8 in revised MS) due to the large local biases present (Figure 4).

*All validation data are within the Canadian domain, but it is not clear that the fit upstream and downstream of these flux increments is equally consistent. This could be easily tested by comparing a forward run of the prior and posterior fluxes to e.g. soundings outside of the optimized domain. If the adjustments lead to a significantly poorer match to these non-assimilated measurements, there may be a problem with this approach. Methane is long-lived, so not only the measurements immediately downwind should be considered. This is the minimum analysis necessary to test if this approach is reasonable.*

This is a reasonable suggestion. A figure has been added comparing the prior and posterior fluxes to NOAA/ESRL observations from surface flasks, aircraft and ship measurements outside of the Canadian domain. This has been added as *Section 1.3.5 Evaluation of the Prior and Posterior Fluxes Using Global Observations Outside of the Canadian Domain*. Figure S8 from this added section is shown below.

The following lines have been added in the Supplement L242-262:

*The inverse model design in this study uses a simplified approach, where Canadian emissions are optimized using only observations in Canada. The results from this approach may be sensitive to errors in the global model projected onto the Canadian domain if errors in the global model are sufficiently large relative to the local biases in Canada (Figure 4 in the main text) and the observational error used in the inversion procedure (16 ppb for GOSAT, 65 ppb for ECCC). Figure S8 shows an independent evaluation of the prior global model and the posterior from this study to 2010–2015 background observations from the NOAA cooperative flask sampling network (https://gml.noaa.gov/ccgg/flask.html) outside of the Canadian domain. We use a simple version of the posterior where Canadian anthropogenic emissions are scaled up by 37% to 6.0 Tg a$^{-1}$ and natural emissions are scaled down by 24% to 11.2 Tg a$^{-1}$. This captures the central results of the monthly, sectoral, and provincial inversions in the main text and avoids a large number of model comparisons. The analysis shows that the prior model reasonably reproduces the methane background, and the posterior from adjusted Canadian emissions does not degrade this result. In the reduced-major axis regression, the prior r$^2$ coefficients are in the range of 0.77–0.92 and the prior slopes are in the range of 0.94–0.97 across the three surface, ship, and aircraft datasets. In the posterior, the r$^2$ is in the range of 0.76–0.91 and the slope is in the range of 0.93–0.96. The posterior reflects a decrease of 2.0 Tg a$^{-1}$ in the global budget due to a net decrease in Canadian emissions, which is shown in the improvements to the mean bias comparisons. This decrease in emissions slightly improves the global model agreement with independent data in the years 2010–2013 (since the model overestimates emissions) and slightly*

*degrades the agreement in 2014–2015 (since the model underestimates tropical emissions), which is understandable considering only Canadian emissions are adjusted and the global model is not optimized. A net decrease in Canadian emissions is consistent with previous global inversion studies using GEOS-Chem (Turner et al., 2015; Maasakkers et al., 2019). The results from the Canada-focused inversion with subnational details in this study show that the net-decrease in Canadian natural emissions masks an increase in anthropogenic emissions in Western Canada which should be considered in global inverse studies.*

[Figure]

**Figure S8:** Model comparison to independent NOAA observations globally from 2010–2015. The top panel shows data used in the global model comparison. Red diamonds indicate NOAA surface flasks, purple circles indicate NOAA ship data, and blue lines indicate HIPPO III, IV and V aircraft data. Comparison of the prior and posterior emissions in GEOS-Chem is shown using a reduced-major axis regression against NOAA Surface flasks (bottom-left), HIPPO III, IV and V aircraft data (bottom-middle), and NOAA Ship data (bottom-right).

*Figure 3, and the discussion of the "acceptable reproducibility of background methane": Again, I am not convinced by this argument that these background fields are so accurate that one can reasonable optimize only Canadian fluxes because everything else agrees so well. Yes, the variability is well matched, but a bias of 5.3 ppb is certainly larger than the measurement uncertainty and, more importantly, looking at the bias alone seems to underestimate the difference. In this plot there seems to be an overestimation of the observations in the earlier part of the record, but by the last year the model seems to be underestimating the measured concentrations. If this boundary condition accepted as is, this difference in the trend will be mapped entirely onto the optimized Canadian fluxes, and not the upstream mismatch where the correction belongs. This trend in the model-observation mismatch appears in Figure 6 as well, even though it is only showing a subset of the simulation period (why?), but at least some of this is already apparent in the un-optimized western boundary. (Notably, the trend of the emissions correction that would be needed to correct this error is positive, just as the trend seen in the ECCC inversion for wetland emissions.)*

While the mean background bias of 5.3 ppb is much larger than the measurement uncertainty for ECCC data, it is much smaller than the observational error used in the inversion of ~65 ppb, which is a combination of instrument error and model error. Hence, the inversion is primarily designed to address large biases observed over Canada and not finer scale characterization.

However, the reviewer is correct that the positive–to–negative change in the background is large enough to influence the trend in the ECCC inversion since the surface data was accepted as is. In the original manuscript, the model background for the surface data was left uncorrected (since surface pixels would be less affected by a model underestimation of tropical emissions) and the model background for the GOSAT XCH$_4$ comparison was corrected. The purpose of presenting the results in this manner was to represent an alternate approach to the bias corrections in the ECCC and GOSAT intercomparison. However, this is confusing since it superimposes two different issues (comparison of ECCC and GOSAT inversion + comparison of using and not using a background correction). This is changed in the revised manuscript and the issue is discussed in better detail.

This analysis is added as *Section 1.3.2 Sensitivity Tests of the ECCC-Constrained Emissions*, shown below are Table S2 and Figure S4 from this analysis. The following lines are added in the Supplement (L111-148):

*While the mean model bias of +5.3 ppb in Figure S3 shows a net over-estimation in the model, the later years 2014 and 2015 show a model underestimation primarily due to underestimated tropical emissions (Maasakkers et al., 2019). This positive-to-negative difference in the model background can project errors onto the trend of ECCC-constrained emissions. This is addressed by removing the annual-mean background bias at the Canadian boundary conditions from the observation vector. We use the westmost boundary condition site ESP and a second northernmost background site at Alert, Nunavut (ALT) to diagnose errors in the methane background and show the annual mean model-observation differences in Table S2. The average of these two sites is used to adjust the model for the base-case ECCC inversion in the main text. In Section 1.3.2 of the Supplement, we test the sensitivity of the posterior emissions to the use of*

*these various background corrections and show consistent results, with the background-adjusted inversion showing slightly more agreement with the GOSAT inversion.*

*Figure S4 shows the sensitivity tests comparing the ECCC inversions with an unadjusted model to the two background-adjusted ECCC inversions using either the mean yearly bias from ESP or ALT. The three inversions are consistent with each other within their error intervals, but the adjusted ECCC inversions show improved agreement with the GOSAT results. For anthropogenic sources, the mean yearly emissions are $6.0 \pm 0.4$ Tg $a^{-1}$ in the unadjusted ECCC inversion, $6.1 \pm 0.4$ Tg $a^{-1}$ with the ESP-adjusted ECCC inversion, and $6.0 \pm 0.4$ Tg $a^{-1}$ with the ALT-adjusted inversion. For natural sources, the mean yearly emissions are $10.5 \pm 1.9$ Tg $a^{-1}$ in the unadjusted ECCC inversion, $12.0 \pm 1.4$ Tg $a^{-1}$ in the ESP-adjusted ECCC inversion, and $11.0 \pm 1.2$ Tg $a^{-1}$ in the ALT-adjusted ECCC inversion. The background-adjusted inversions show higher natural emissions compared to the unadjusted case in the years 2010–2014, and lower natural emissions in 2015 due to the negative background bias that is removed. The background-adjusted inversions show better agreement with the GOSAT mean yearly natural emissions of $11.7 \pm 1.2$ Tg $a^{-1}$. In addition, the trend in natural emissions over this time period is reduced by 40-45% from 1.0 Tg $a^{-1}$ in the unadjusted inversion to 0.55–0.60 Tg $a^{-1}$ in the adjusted inversions. These results show that the background error does not largely affect the average 2010–2015 results regarding the overall increase in anthropogenic emissions and decrease in natural emissions. Correcting for the model background minimizes the projection of under-estimated tropical emissions onto the Canadian fluxes in the later years, which improves the consistency with the GOSAT inversion and significantly reduces the presence of a large trend that was not corroborated by GOSAT.*

**Table S2:** *Mean annual model-measurement differences at background sites ESP and ALT.*

| | Mean Model–Measurement Difference (ppb) | | |
|---|---|---|---|
| *Year* | *ESP[a]* | *ALT[b]* | *Average[c]* |
| *2010* | *+5.0* | *+8.8* | *+6.9* |
| *2011* | *+5.8* | *+8.5* | *+7.2* |
| *2012* | *+3.6* | *+5.9* | *+4.8* |
| *2013* | *+2.6* | *+10.5* | *+6.6* |
| *2014* | *+2.1* | *+11.3* | *+6.7* |
| *2015* | *–6.9* | *–4.7* | *–5.8* |

*[a]Site ESP is located at 49.38°N, 126.54°W, and is the westernmost boundary condition for Canada.*
*[b]Site ALT is located at 82.45°N, 62.51°W, and is the northernmost boundary condition for Canada.*
*[c]The average is used in the base-case ECCC inversions shown in the main text. The three alternatives: adjustments using ESP, ALT and no background adjustments are shown as sensitivity tests in the Supplement.*

Considering this background-adjusted analysis reduces the effect of global model errors, it is more suitable to use the mean of the background-adjusted ECCC inversions as the base case in the main text (referred to as "Posterior ECCC" in the revised manuscript), and move the unadjusted ECCC inversion as a senstivity test in the Supplement. This is reflected in minor changes to Figure 5, 6, 7, 8 (which are the old Figures 7-10). In the Supplement, this change is reflected in Figure S2, Table S4 and Table S5 with the analyses re-done using the background-adjusted ECCC inversion as the base case.

[Figure]

**Figure S4**: Sensitivity analysis of inversion results depending on the use of model background correction for surface pixels. Referred to as the monthly inversion, this approach optimizes annual total Canadian anthropogenic emissions (top) and monthly total natural emissions (bottom) in an n = 78 state-vector element setup. The prior emissions (gray) are compared to the posterior results using GOSAT (green), and the posterior using ECCC data with an unadjusted

background (blue), ECCC data using a background adjusted according to the yearly difference at ESP (teal) and ALT (purple) from Table S2.

*Furthermore, the argument that Maasakkers et al. (2020) showed "relatively minimal" adjustments to US emissions near the Canadian border does not mean that US fluxes from further afield do not affect concentrations measured in Canada. Yes, the winds are generally westerly, but air certainly crosses the border in both directions. Not to mention that the stations Egbert and Sable Island have a great deal of US signal when considering only westerly flow, as they are well south of the 49th parallel.*

To address the concern regarding the influence of US emissions near the Canadian border, we introduce a sensitivity test where these two stations most influenced by cross-border transport, Egbert (EGB) and Sable Island (SBL) are removed from the ECCC inversion. Figure S5 shows a sensitivity test where EGB and SBL (at latitudes of 44.2°N and 43.9°N, respectively) are removed (note in this case, the background is left un-adjusted to avoid overlap in the issues). The mean of anthropogenic emissions in the inversion without these stations is $6.4 \pm 0.6$ Tg a$^{-1}$, and the mean of natural emissions is $10.9 \pm 1.5$ Tg a$^{-1}$. These results are similar to the posterior from the unadjusted ECCC inversion ($6.0 \pm 0.4$ Tg a$^{-1}$ anthropogenic, $10.5 \pm 1.9$ Tg a$^{-1}$ natural) and the GOSAT inversion ($6.5 \pm 0.7$ Tg a$^{-1}$ anthropogenic, $11.7 \pm 1.2$ Tg a$^{-1}$ natural). This sensitivity test shows that the US signal has a minimal influence on the optimization of the large biases due to Canadian emissions. This sensitivity test is added to Section 1.3.2 of the Supplement.

Figure S5 is added and the following new text in the Supplement (L150-165):

*To address the possibility of US emissions influencing the posterior results near the Canadian border, we show a sensitivity test where the two stations most influenced by cross-border transport, Egbert (EGB) and Sable Island (SBL) are removed from the ECCC inversion. Figure S5 shows posterior-ECCC emissions where EGB and SBL (at latitudes of 44.2°N and 43.9°N, respectively) are removed (note in this case, the background is left un-adjusted to avoid overlap in the issues). The mean of anthropogenic emissions in the inversion without these stations is 6.4 ± 0.6 Tg a$^{-1}$, and the mean of natural emissions is 10.9 ± 1.5 Tg a$^{-1}$. These results are similar to the posterior from the unadjusted ECCC inversion (6.0 ± 0.4 Tg a$^{-1}$ anthropogenic, 10.5 ± 1.9 Tg a$^{-1}$ natural) and the GOSAT inversion (6.5 ± 0.7 Tg a$^{-1}$ anthropogenic, 11.7 ± 1.2 Tg a$^{-1}$ natural). This sensitivity test shows that the US signal does not substantially affect the results from the optimization of large biases observed by Canadian observations due to Canadian emissions.*

[Figure]

**Figure S5**: Sensitivity analysis of inversion results depending on the inclusion of sites EGB and SBL which are sensitive to cross-border transport from the United States. Similar to Fig. S4, the monthly inversion optimizes annual total Canadian anthropogenic emissions (top) and monthly total natural emissions (bottom) in an n = 78 state-vector element setup. The prior emissions (gray) are compared to the posterior results using GOSAT (green), and the posterior using ECCC data including all sites (blue) and ECCC data excluding EGB and SBL (yellow).

*The suggested increase in biogenic fluxes from 2010-2015 from the in-situ network is massive –
this is on the order of 10% per year! This would be an extraordinary finding, if it can be
substantiated. How might this be tested? Did you consider looking at isotope measurements, for
example? Why might this not be seen in the GOSAT-only inversion? Why were the GOSAT and
ECCC measurements not combined in this "standard" inversion setup as well (as they were in
the "policy-themed" inversions presented in Section 3.4). It seems an obvious natural step to do
so, to see if this trend is still apparent.*

In the original text it was stated

*L509-L511: The lack of corroboration of trends between ECCC and GOSAT data may be
reflective of the lower overall sensitivity of total column methane to these surface fluxes (Sheng
et al., 2017; Lu et al., 2020) or the inability of this inverse system to constrain trends sufficiently.*

Given the above results of the sensitivity tests and the reduction in the trend by 40–45% by using
a background correction, the latter part of this statement accounts significantly for this result.
This entire section has been reworded to better communicate the influence of the background
corrections on the magnitude of the trend. In general, the limitations of the method and the study
period of 6 years is insufficient for a complete discussion of trends, and it is not a central focus
of this study, but the presented trends from ECCC and GOSAT show better agreement when the
background bias corrections are similarly matched.

The ECCC and GOSAT data were not combined in this monthly inversion setup to show a
comparison of results from the two datasets. A combined ECCC+GOSAT monthly inversion is
added and shown in the Supplement, which is within the two results.

The following lines (L454-455) have been changed:

*a linear fit to the posterior annual emissions using ECCC data shows a trend of increasing
natural emissions at a rate of ~1.0 Tg a$^{-1}$ per year from 2010–2015*

To:

*a linear fit to the posterior annual emissions using ECCC data shows a trend of increasing
natural emissions at a rate of ~0.56 Tg a$^{-1}$ per year from 2010–2015*

to reflect the change in the base case to the background-adjusted ECCC inversion.

The following lines have been added regarding the discussion of trends (L459-465):

*The combined ECCC+GOSAT inversion using this setup is consistent with the results of the
individual inversions, it is shown in the Supplement (Fig S11) while the intercomparison is
emphasized here, although we note the combined inversion also does not corroborate this trend.
We evaluate the possible influence of errors in the global model on the projection of a trend onto
the ECCC inversion in Section 1.3.2 of the Supplement. While the mean natural emissions over
2010–2015 show consistent results in the sensitivity tests, the limitations of the observation*

*system, the inversion procedure and the timescale of the analysis limit the interpretation of trends.*

*Once these concerns have been addressed the study would be appropriate for publication, but until the robustness of this "regional adjustment only" approach has been tested against independent measurements upwind and downwind of Canada in forward runs of both the prior and posterior fluxes, the scientific conclusions cannot be considered sufficiently robust.*

Thank you for the helpful and clear feedback to improve the quality of the manuscript.

**Minor/typographic comments:**

*L17: have been conflicting -> conflict*
*L26: slower -> a slower*
*L35: specify anthropogenically-influenced GHG: CO2 is less significant than H2O…*
*L54: because only a 3% source-sink imbalance, -> because a source-sink imbalance of only 3%,*

These lines have been changed as suggested.

*L58: Please specify that the "Canadian greenhouse gas inventory" is not just an inventory of some prior integrated over Canada, but rather the government report of emissions submitted to the UNFCCC. This is a bit confusing. It's mentioned in the abstract and fully capitalized, as if it were the proper name: "the Canadian Greenhouse Gas Inventory". But then it's also called the "National GHG Inventory" (also capitalized, also in the abstract), and then here just "the Canadian greenhouse gas inventory". None of these match the title of the actual document, which should be explicitly introduced in the introduction.*

The document has now been explicitly introduced as the National Inventory Report: Greenhouse Gas Sources and Sinks in Canada, a government report of Canada's emissions submitted to the United Nations Framework Convention on Climate Change (UNFCCC) in the abstract (L14-15) and in the introduction (L46-47). It is thereafter referred to as the National Inventory.

*L73: compromising interpretation -> compromising the interpretation*
*L83: wetlands fluxes -> wetland fluxes*
*L86-87: „an increase in" and "a decrease in" would be clearer than "upscaling" and "downscaling" in this context, which could be interpreted as spatial extrapolation/(dis)aggregation.*
*L93: insert comma after first use of "emissions"*
*L113: insert commas before and after "Estevan Point (ESP)"*
*L126: mol -> mole*
*L127: local time for when -> local time, when the*
*L129: western most -> westernmost*
*L136: I guess you mean the largest methane fluxes from wetlands in North America?*

These corrections have been added.

*Section 2.2: I was surprised to see biomass burning not mentioned explicitly in the text, but only listed in the table. It can have quite a bit of interannual and regional variability. I was also surprised to see that the termite emissions were identical to those of biomass burning (in Canada??), and also geologic seeps. Is this just a coincidence, or were these three small sources just distributed evenly over the three (rather different) prior spatial distributions? Please clarify this, also in the text.*

Biomass burning is heavily aliased by wetlands emissions in this inversion setup, and the observation network is not capable of resolving the two methane sources. Emissions for biomass burning is from QFED (Darmenov and da Silva, 2013) and termite emissions are from Fung et al. (1991). Seeps and other global sources are described in Maasakkers et al. (2019). These are different inventories with different spatial patterns, the magnitudes being similar over Canada is by coincidence.

The following line has been added (L185-188): *Natural emissions are divided into wetlands, which are 14.0 Tg a$^{-1}$ in the ensemble mean, and other natural emissions, which are 0.8 Tg a$^{-1}$ from biomass burning, seeps, and termites. Each component of other natural emissions has a separate spatially disaggregated inventory as described in Maasakkers et al. (2019).*

*L191: A couple concerns here, one minor and one major. Here it is optimistically stated that the spatial pattern of emissions "may" show less agreement: this is almost certainly the case, just from a statistical perspective. The major concern has to do with the use of a global simulation where only the Canadian emissions are allowed to be optimized, and where only measurements over Canada are assimilated, but this is discussed elsewhere.*

L184-185 now says: *however we cannot compare the spatial pattern of emissions which will likely show more discrepancies*. The major concern is addressed in the previous revisions.

*Figure 2: it seems a mistake that the contiguous US/Greenland is not screened out in panel D (but Alaska is), while it is for the other three panels.*

Fixed in Figure 2.

*L250: remove "done"*
*L259: needs a connecter after the comma (e.g. "such that the", "wherein", "and" ...)*
*L281: insert "and" before "other"*
*L293: space missing?*

These corrections have been added.

*L294: Did I understand correctly that the in situ data were averaged over the local afternoon each day, essentially giving just one data point per day per station (as described in line 127)? If so, a mean observational error of 65 ppb seems rather massive! Can this be attributed to a poor representation of the spatial distribution of the fluxes, which is not optimized explicitly? The only*

*way the model can adjust the spatial distribution is by changing the weighting of the various categories.*

Yes, the ECCC data amounts to one data point per day per station. The observational error of 65 ppb is much larger than the instrument error due to model representation error using the 2°x2.5° grid. The goal in this study is to address large-signal biases using a relatively sparse observation system, so this broad-category approach is useful. In future studies, model representation error and the characterization of observational error correlations can be improved to meet a goal of optimizing finer scale emissions using a superior observational network.

*L374, L376 (and elsewhere – find and replace): change "wetlands emissions" to "wetland emissions"*
*L396: and compares -> and compares them*
*L400: from region -> either "from regions" or "from the region"*

These corrections have been added.

*L406: Is Egbert really sensitive to emissions from the Hudson Bay Lowlands? This surprises me. Fraserdale, sure, and maybe Chibougamau, but Egbert? Out of curiosity: for the simulations shown in Figure 6, were the different WetCHARTS scenarios also used for the US fluxes, or were these fixed? Also, please specify that the anthropogenic and "other natural" fluxes from Table 2 were used in the forward simulations shown in Figure 6 (which I assume to be the case).*

We are able to test this using the output from the tagged tracer simulation. We calculate the maximum ΔCH4 to be ~100 ppb each year in the summer specifically from Canadian wetlands emissions when running the simulation with the mean of WetCHARTS scenarios. This matches the mean of the summertime peaks shown in Figure 6; hence the seasonal pattern is primarily accounted for from the ΔCH4 due to Canadian wetlands and not US wetlands. The different WetCHARTS scenarios shown in Figure 6 are not limited to Canada, so the change caused by variable US wetlands emissions is also included in this figure. We show in the previous sensitivity test that excluding EGB and SBL from the ECCC inversion results in similar posterior emissions, so the effect from the US on the results is minimal.

The caption in Figure 3 is changed to specify "…differing in the use of WetCHARTS ensemble members for wetland emissions, with other emissions corresponding to Table 2.".

*L588-590: I don't understand this sentence entirely, there seems to be words missing. Perhaps you mean: While there are about 5 times more GOSAT observations than ECCC observations for use in the analysis and the in-situ observations have larger observational error in Sa (due to model error), the surface measurements are much more sensitive to surface fluxes, which offsets the weight of the larger amount of GOSAT data. Or something like that?*

This line has been changed to the suggested: "While there are about 5 times more GOSAT observations than ECCC observations for use in the analysis and the in-situ observations have larger observational error in Sa (due to model error), the surface measurements are much more sensitive to surface fluxes, which offsets the weight of the larger amount of GOSAT data".

*L688: should "or" be "and"?*

Corrected.

*In Supplement:*
*P7 L167: out -> our*

Corrected.

*Figures S4 and S5: I wonder if these figures might not be easier for the reader to interpret if they were presented as matrices/surface plots? The amplitude of e.g. the singular vector decomposition in the bottom plots could still be indicated somehow, or even kept as line figures, which would help avoid confusion about the interpretation of dashed lines in the middle and bottom panels of Figure S5.*

Added more reader-friendly figures using the suggested surface plots.

**Reviewer 2**

**Overall suggestions:**

*I felt that the article introduction could flow better with better connection/continuity among topics. Some of the information in the introduction felt out of place (see specific suggestions below). I would think about what argument you want to make in each paragraph of the introduction and use informative topic sentences to guide the reader through each of these arguments.*

The introduction has been condensed and L47–56 on the discussion of trends has been removed.

*It seems like background estimation was a difficult and challenging process in this study. I would consider adding a second approach to estimating the background -- either by optimizing global fluxes as part of the inverse model or by using a background constructed using atmospheric observations (instead of a model-based background). Section 3.1 of the manuscript includes a lengthy discussion of the merits of the model-based background and whether it is sufficiently accurate for the task at hand. Instead of this lengthy discussion, a second background estimate might be a better way to succinctly quantify the impacts of background uncertainties on the estimated methane fluxes.*

We added the use of an alternate observation-adjusted background for the ECCC inversion (discussed in Reviewer 1 comments) and included several new sensitivity tests in the Supplement 1.3 to test the use of the observation background (Fig S4) and to test the influence of nearby US emissions (Fig S5). The comparisons better quantify the impacts of background uncertainties and show consistent results.

*Several sections of the manuscript are relatively long and wordy, especially section 3. In many cases, I think you could cut or condense some of the written material to yield a leaner, punchier, more concise manuscript.*

*Section 3.1* has been expanded to address Reviewer 1 comments and moved to the Supplement. This simplifies Section 3 to be more concise following the suggestion of Reviewer 2.

*I also have some concerns about the inverse modeling setup. I understand that redesigning the inverse modeling framework would require large numbers of new GEOS-Chem runs; hence, I would strongly urge the authors to revise their inverse modeling setup for future studies (even if not the current study). The inverse modeling simulations used in this study either (1) optimize the temporal distribution of fluxes assuming the spatial distribution of the prior is correct, or (2) optimize the spatial distribution of fluxes assuming that the temporal distribution of the prior is correct. In reality, I think both the spatial and temporal distribution of the prior flux estimate could be improved through inverse modeling, and it would be ideal to design an inverse model that does both. Otherwise, I worry that errors in one could interfere your inferences about the other. Also, I think you would see higher model-data correlations in Fig. 10 if your inverse models had more flexibility to adjust both the spatial and temporal distribution of emissions. In*

*future studies, I would also consider using nested North America GEOS-Chem runs instead of using much coarser 2 x 2.5 resolution global simulations.*

This is a reasonable suggestion that has promise, however in this study we show that the design of the inverse model is primarily limited by the observation network. For example, in *Section 3.3 Joint-inversions Combining ECCC In situ and GOSAT Satellite Data,* it was shown that even large spatially aggregated neighboring provinces could not be properly distinguished. We agree a superior design of the inverse model using GEOS-Chem would be to optimize emissions using a nested grid simulation to invert emissions at better spatial and temporal resolution. However, we feel a more sophisticated model would better suit a more sophisticated observation system. Improvements to the observation network include an expanded ECCC surface network and satellite observations with either higher density (TROPOMI) or higher precision (GOSAT–2) outside of the years of this analysis.

The following lines (L296-309) have been added to the end of the methodology to better clarify the limitations of the study and the specific objectives:
* * *
*In summary, the inverse model is designed to suit the objectives of this study, which are to: (1) optimize anthropogenic and natural emissions in Canada at the national-scale and (2) compare the results of inversions using surface and satellite observations, and (3) characterize the limitations of the observing system towards subnational-scale emissions discretization. The spatial and temporal resolution of the inversion is limited by the precision of GOSAT data, the precision of the model representation of surface methane for ECCC data, and the sparse coverage of both systems relative to the smaller magnitude of Canadian emissions. This simplified approach, where Canadian emissions are optimized using only observations in Canada, may be sensitive to errors in the global model that are projected onto the Canadian domain. This is minimized if errors in the regional representation of methane, which are corrected in the inversion, are much larger than errors in the background from the global model, or if the background methane is corrected using global observations outside of the Canadian domain. We show an analysis of the global model alongside sensitivity tests of the inversions in Section 1.3 of the Supplement which produce consistent results. Future studies may deploy a more sophisticated, high resolution inverse model that will match more sophisticated observations, which include an expanded ECCC surface network, as well as satellites with higher density (TROPOMI; Hu et al., 2018) or higher precision (GOSAT-2; Nakajima et al., 2017) observations outside of the years of this analysis.*

**Specific suggestions:**

The abstract is very long at about 400 words. I would consider making the abstract punchier and more concise.

The abstract has been condensed from 408 to 367 words.

Line 37 "however recent trends in": I think this phrase should be a separate sentence from the previous sentence.

Corrected to a new sentence.

Line 47 - 56: This paragraph feels out of place. It doesn't flow with the previous or subsequent paragraphs, and it is not clear how global atmospheric trends are relevant to the current study on Canada. I would consider cutting this paragraph.

This paragraph on the discussion of trends has been removed.

Line 58: I wasn't clear how the first and second sentences of this paragraph relate to one another. I would try to find a topic sentence for this paragraph that summarizes the overall objective of this paragraph. You might want to have one paragraph about Canada's emissions inventory and another paragraph about existing studies instead of putting both topics in the same paragraph.

This section has been changed to the following two paragraphs (L46-63):

*In the Government of Canada's submission to the United Nations Framework Convention on Climate Change (UNFCCC), hereafter referred to as the National Inventory, anthropogenic emissions are estimated to be 4.1 Tg $a^{-1}$ in 2015, with 68% of emissions originating from the Western Canadian provinces of Alberta (42%), Saskatchewan (17%) and British Columbia (9%). Sectoral contributions over the entire country are from three categories: Energy (49%), Agriculture (29%) and Waste (22%) (Environment and Climate Change Canada, 2017). Natural emissions, which are mostly due to Boreal wetlands, are highly uncertain, on the order of ~10-30 Tg $a^{-1}$ from biosphere process modelling (Miller et al., 2014; Bloom et al., 2017).*

*Atmospheric observations provide constraints on methane emissions. Studies constraining anthropogenic and/or natural methane emissions within Canada have included the use of surface in situ measurements (Miller et al., 2016; Atherton et al., 2017; Ishiziwa et al., 2019), aircraft campaigns (Johnson et al., 2017; Baray et al., 2018) and satellites (Wecht et al., 2014; Turner et al., 2015; Maasakkers et al., 2021). These observations can determine emissions through mass balance methods or be used in conjunction with a chemical transport model (CTM). Bayesian inverse modelling constrains prior knowledge of emissions based on the mismatch between modelled and observed concentrations. This requires reliable mapping of "bottom-up" inventory emissions for the "top-down" observational constraints to be useful (Jacob et al., 2016). Inverse modelling has been more challenging for Canada than the United States due to a) the sparsity of surface stations and satellite data (Sheng et al., 2018a), b) a factor of ~10 lower anthropogenic emissions (Maasakkers et al., 2019), c) large spatially-overlapping emissions from Boreal wetlands that are highly uncertain (Miller et al., 2014), and d) model biases in the high-latitudes stratosphere (Patra et al., 2011), compromising the interpretation of observed methane columns.*

Line 75 "however studies have": I think this should be the beginning of a new sentence.

This line has been changed to two sentences: *These observing system challenges have made Canadian methane emissions difficult to quantify. However, studies show a consistent story across different scales and measurement platforms.*

Line 75 "have been showing": replace with "show"

Corrected.

Lines 75 - 95: The information in this paragraph overlaps with the information in the previous paragraph. I would either come up with a unique topic sentence for this paragraph to differentiate this paragraph from the previous one, or I would combine the discussion of top-down studies in this paragraph with the discussion of top-down studies in the previous paragraph.

To differentiate the first methodology paragraph with the second results paragraph, the following changes have been made:

The first paragraph is differentiated by beginning with (L53): *Atmospheric observations provide constraints on methane emissions.* The second paragraph begins with (L65): *These observing system challenges have made Canadian methane emissions difficult to quantify. However, studies show a consistent story across different scales and measurement platforms*.

Line 105 "intercomparison": Why not use "comparison" instead?

Changed.

Line 116 "mean along other GEOS-Chem prior emissions": It feels like there is a word missing here.

Changed L107 to *mean in addition to other GEOS-Chem prior emissions*.

Line 147: Was the Chibougamau site decommissioned in 2011, or did it come back online into operation after 2015? This distinction isn't clear in the wording of line 147.

Adjusted wording. The Chibougamau site was moved to an alternate location with a new name. This line now reads: *Chibougamau, Quebec is replaced by Chapais, Quebec ~50 km away from 2011 onwards, overlapping in Fig.1*.

Figure 1: I believe that ECCC has several observation sites in Northwest Territories and Nunavut. Why not include those sites in the inverse model? See the list of ECCC sites shown in Fig. 2.6: https://www.nrcan.gc.ca/sites/www.nrcan.gc.ca/files/energy/Climate-change/pdf/CCCR_FULLREPORT-EN-FINAL.pdf.

The sites included in this study were those that were made available on the public domain for the scientific community on the World Data Centre for Greenhouse Gases (WDCGG; https://gaw.kishou.go.jp/). Measurements that were not yet available publicly could not be used.

Equations 1, 2, and 3: I think that vectors should be displayed in bold-italic font and matrices in bold font.

Corrected the fonts.

Line 235: Should the dimensions of K be m by n, given the definitions for m and n in the article?

Corrected to *m by n*.

Line 248: What are you optimizing for in the monthly inversion? Are you estimating methane fluxes from each individual model grid box in each month? If that were the case, I think the value of n here would be larger. Or are you optimizing something else?

The inversion is not by grid box due to limitations in the observation network. We are optimizing all aggregated Canadian anthropogenic emissions according to a yearly scaling factor, and all aggregated Canadian natural emissions according to monthly scaling factors. These limitations in the state vector *n* are ultimately due to the limitations of the surface and satellite observation network. We explore the limitations of in the combined inversions Section 3.3 and Supplement S1.4.

Line 252: I wouldn't refer to a monthly inversion as "high temporal resolution". I have seen existing studies estimate daily methane fluxes in an inverse model, and numerous inverse modeling studies of CO2 estimate 3-hourly fluxes.

Corrected. This line is rephrased to: *The monthly inversion provides higher temporal resolution relative to the other approaches in this study to constrain the seasonality of natural emissions, assuming the spatial distribution is correct*.

Line 474 - 475: I disagree that there's a tradeoff between spatial resolution and temporal resolution in the inverse model. Alternative approaches would be to (1) use the GEOS-Chem adjoint in the inverse model, or (2) use a Lagrangian model like Flexpart or STILT in the inverse model. Those approaches would not necessitate a trade-off between the spatial and temporal resolution of the inverse model.

The model itself has the technical capabilities to resolve emissions both spatially and temporally. For the Canadian domain, the limitations are not due to the model but due to the observation network, and the design of the coarse inverse model used in this study is chosen to suit this problem. This line is changed to: *This approach is a trade-off of time for space, due to the limitations of the observations, giving up finer spatial resolution for finer temporal resolution*.

Line 582 "magnitude emissions in Canada": Is there a word missing here?

Changed to: …*constraining emissions in Canada with very small magnitudes*.

We thank Reviewer 2 for their time. The suggestions have improved the quality of the manuscript.

**References**

Darmenov, A. and da Silva, A.: The quick fire emissions dataset (QFED)–documentation of versions 2.1, 2.2 and 2.4, NASA Technical Report Series on Global Modeling and Data Assimilation, NASA TM-2013-104606, 32, 183 pp., 2013.

Fung, I., John, J., Lerner, J., Matthews, E., Prather, M., Steele, L., and Fraser, P.: Three-dimensional model synthesis of the global methane cycle, J. Geophys. Res.-Atmos., 96, 13033–13065, 1991.

Maasakkers, J. D., Jacob, D. J., Sulprizio, M. P., Scarpelli, T. R., Nesser, H., Sheng, J.-X., Zhang, Y., Hersher, M., Bloom, A. A., Bowman, K. W., Worden, J. R., Janssens-Maenhout, G. and Parker, R. J.: Global distribution of methane emissions, emission trends, and OH concentrations and trends inferred from an inversion of GOSAT satellite data for 2010–2015, Atmos. Chem. Phys., 19(11), 7859–7881, doi:10.5194/acp-19-7859-2019, 2019.

Maasakkers, J. D., Jacob, D. J., Sulprizio, M. P., Scarpelli, T. R., Nesser, H., Sheng, J., Zhang, Y., Lu, X., Bloom, A. A., Bowman, K. W., Worden, J. R., and Parker, R. J.: 2010–2015 North American methane emissions, sectoral contributions, and trends: a high-resolution inversion of GOSAT observations of atmospheric methane, Atmos. Chem. Phys., 21, 4339–4356, https://doi.org/10.5194/acp-21-4339-2021, 2021.

Stanevich, I., Jones, D. B. A., Strong, K., Parker, R. J., Boesch, H., Wunch, D., Notholt, J., Petri, C., Warneke, T., Sussmann, R., Schneider, M., Hase, F., Kivi, R., Deutscher, N. M., Velazco, V. A., Walker, K. A., and Deng, F.: Characterizing model errors in chemical transport modeling of methane: impact of model resolution in versions v9-02 of GEOS-Chem and v35j of its adjoint model, Geosci. Model Dev., 13, 3839–3862, https://doi.org/10.5194/gmd-13-3839-2020, 2020.

Turner, A. J., Jacob, D. J., Wecht, K. J., Maasakkers, J. D., Lundgren, E., Andrews, A. E., Biraud, S. C., Boesch, H., Bowman, K. W., Deutscher, N. M., Dubey, M. K., Griffith, D. W. T., Hase, F., Kuze, A., Notholt, J., Ohyama, H., Parker, R., Payne, V. H., Sussmann, R., Sweeney, C., Velazco, V. A., Warneke, T., Wennberg, P. O. and Wunch, D.: Estimating global and North American methane emissions with high spatial resolution using GOSAT satellite data, Atmos. Chem. Phys., 15(12), 7049–7069, doi:10.5194/acp-15-7049-2015, 2015.

---

## Author Response (AR2)

Response to Editor

Thank-you for your time. We have responded to the final technical suggestions by the reviewer comments below. We have not changed the main paper, other than one minor typo. The technical changes are all in the Supplement.

Reviewer Comments

Thank you for your responses to the reviewer comments, most of the questions have been appropriately addressed. I think that the analysis of potential biases in the inflow/background methane mixing ratios has improved the study significantly. One slight complaint is the assessment of the prior and posterior fluxes outside of the domain, illustrated in Figure S8: Here it may have been more informative to consider measurement sites at the same latitude band separately to better see the impact on the outflow, rather than combining all (non-Canadian) global data, or perhaps to have considered the mean bias as a function of time for a subset of data most likely to be affected. But this is rather a suggestion for future studies: as it stands now, the paper is appropriate for publication one the labelling issues described below have been addressed.

Technical suggestions: In the supplemental figures, the plots should be clearly labelled to indicate if they are showing anthropogenic fluxes (only), natural fluxes (only), or total methane emissions. While this is explained in the caption for Figures S4 and S5, it should be on the axis label, or otherwise marked on the figure (as in Figure S11). Likewise, I believe that Figures S2 and S7 are showing natural fluxes only, but this is indicated neither in the figure labels nor in the caption. Please amend this.

Response: We thank the reviewer for their time, and will consider these comments for future studies.

Yes, we agree with the reviewer that the figures should be labelled to add clarity. We have added appropriate labels (either Anthropogenic Methane Emissions or Natural Methane Emissions) to all appropriate Figures including Figures S2, Figure S4 a, b, Figure S7 and Figures 11, a, b.

Robert McLaren & Sabour Baray